# Do Vision-Language Pretrained Models Learn Composable Primitive Concepts?

**Tian Yun**                                                            *tian_yun@brown.edu*
*Department of Computer Science*
*Brown University*

**Usha Bhalla**                                                        *usha_bhalla@brown.edu*
*Department of Computer Science*
*Brown University*

**Ellie Pavlick**                                                      *ellie_pavlick@brown.edu*
*Department of Computer Science*
*Brown University*

**Chen Sun**                                                                *chensun@brown.edu*
*Department of Computer Science*
*Brown University*

**Reviewed on OpenReview:** *https://openreview.net/forum?id=YwNrPLjHSL*

## Abstract

Vision-language (VL) pretrained models have achieved impressive performance on multi-modal reasoning and zero-shot recognition tasks. Many of these VL models are pretrained on unlabeled image and caption pairs from the internet. In this paper, we study whether representations of *primitive* concepts–such as colors, shapes, or the attributes of object parts–emerge automatically within these pretrained VL models. We propose a two-step framework, Compositional Concept Mapping (CompMap), to investigate this. CompMap first asks a VL model to generate concept activations with text prompts from a predefined list of primitive concepts, and then learns to construct an explicit composition model that maps the primitive concept activations (e.g. the likelihood of *black tail* or *red wing*) to composite concepts (e.g. a *red-winged blackbird*). We demonstrate that a composition model can be designed as a set operation, and show that a composition model is straightforward for machines to learn from ground truth primitive concepts (as a linear classifier). We thus hypothesize that if primitive concepts indeed emerge in a VL pretrained model, its primitive concept activations can be used to learn a composition model similar to the one designed by experts. We propose a quantitative metric to measure the degree of similarity, and refer to the metric as the *interpretability* of the VL models' learned primitive concept representations. We also measure the classification accuracy when using the primitive concept activations and the learned composition model to predict the composite concepts, and refer to it as the *usefulness* metric. Our study reveals that state-of-the-art VL pretrained models learn primitive concepts that are highly useful for fine-grained visual recognition on the CUB dataset, and compositional generalization tasks on the MIT-States dataset. However, we observe that the learned composition models have low *interpretability* in our qualitative analyses. Our results reveal the limitations of existing VL models, and the necessity of pretraining objectives that encourage the acquisition of primitive concepts.[1] [2]

---

[1]Code is available at: `https://github.com/tttyuntian/vlm_primitive_concepts`
[2]Project homepage: `https://vlm-primitive-concepts.github.io/`

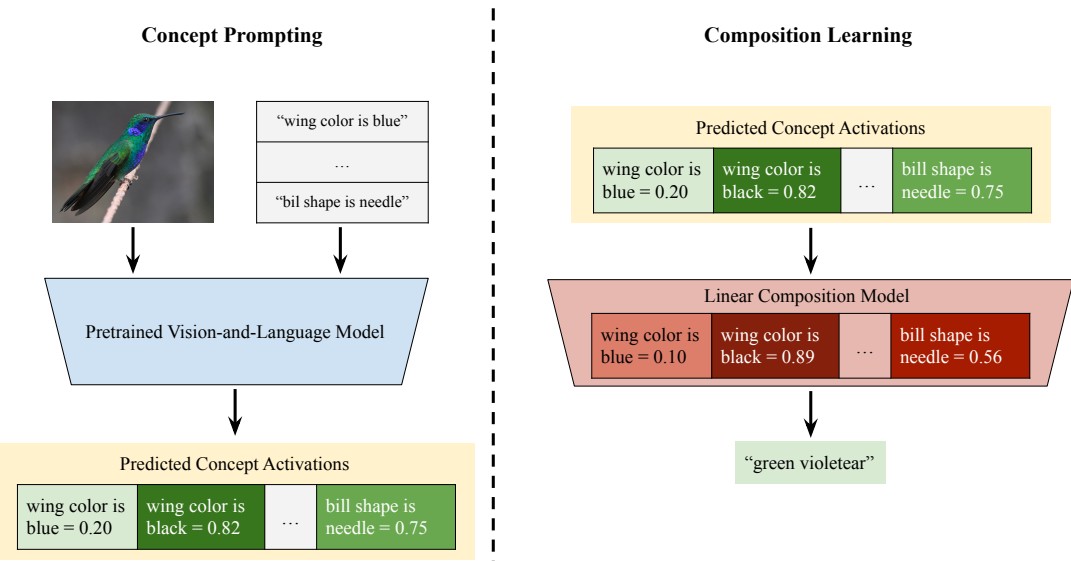

Figure 1: Illustration of our two-step framework, Compositional Concept Mapping (CompMap). In "Concept Prompting" module, a pretrained vision-and-language (VL) model is used to generate concept activations from images and a set of concepts, where each concept is represented by one or a few text prompts. In "Composition Learning" module, a linear composition model is trained on concept activations (either predicted by a pretrained VL model or annotated by human experts) to predict the composite concepts (e.g. *green violetear*) with supervised learning.

## 1    Introduction

Vision-language (VL) models pretrained on raw images and text (Radford et al., 2021; Lu et al., 2019; Sun et al., 2019) have revolutionized deep learning in the last few years. They have demonstrated strong transfer learning performance on a wide range of tasks (Ramesh et al., 2021), even under the zero-shot setup, where text prompts (Brown et al., 2020) are used to specify a task instead of labeled data. Many of these VL models are pretrained on naturally labeled data, such as image and caption pairs from the internet, and thus have the potential to encode commonsense knowledge by processing huge quantities of multimodal data. These models often learn to label complex concepts (e.g., an *golden delicious apple*, or a *bohemian kingfisher*) impressively well. However, it is not clear whether they do this by *implicitly* learning to reason over lower-level primitive concepts (e.g., the color, shape, and texture of an apple, or the *wing color* or *bill shape* of a bird) that humans naturally use to characterize these complex concepts. In this paper, we ask whether pretrained VL models capture representations of such primitive concepts "for free" in the course of their pretraining. If they do so, it has important implications for the capacity of models to support compositional generalization, and for humans to interpret the reasoning procedures models undertake.

Why are we interested in the primitive concepts if a pretrained VL model can directly recognize composite concepts via prompts? For one reason, primitive concepts provide greater interpretability of and interaction with models (Ghorbani et al., 2019; Mao et al., 2019). They can serve as concept bottlenecks (Koh et al., 2020), which allow users to inspect which concepts contribute the most to a prediction or to correct model behaviors by correcting mistakes on the primitive concepts. An interpretable notion of primitive concepts also allows users to teach new concepts efficiently, by specifying how a new concept (e.g. a *red apple*) can be derived from primitives (e.g. color *red* and object *apple*). Existing systems for these applications require manual annotation of the primitives to build corresponding classifiers (Lampert et al., 2009). Therefore, it would be particularly appealing if pretrained VL models are able to learn the primitives automatically.

In this paper, we propose a computational framework, Compositional Concept Mapping (CompMap), to quantify the extent to which a VL model has learned representations of primitive concepts. We focus on

composite concepts which can be unambiguously[3] described using a collection of non-overlapping primitive concepts[4]. The key motivation of our proposed framework is that this description can be obtained in a data-driven fashion, by learning a *linear* classifier from the primitive concepts to the composite concepts. We refer to this linear classifier as a *composition model*. When the true primitive and composite concept annotations are available for a dataset, one can reliably learn a composition model, which we validate experimentally. To align with the more common situation when the primitive concept annotations are not available, we ask a pretrained VL model to automatically annotate the primitive concepts via its *natural language* prompts (Radford et al., 2021). Intuitively, if a VL pretrained model does learn primitive concepts, its concept annotations should allow us to learn the same or a very similar composition model. An illustration can be found in Figure 1.

We quantify this with two metrics. First, we measure the *usefulness* of a learned composition model (e.g., trained with *red apple* and *yellow banana*) for recognizing composite concepts (*red banana*), in terms of its ability to classify compositions of primitive concepts (*red*, *banana*). Second, we evaluate the *interpretability* of a learned composition model. We build on our intuition above and make the assumption that, if a VL pretrained model's primitive concept predictions are accurate and thus interpretable, it should allow the learning of a composition model that generalizes well to the ground truth primitive concepts (e.g. provided by a human expert). Thus, we learn a "ground truth composition model" when given true primitive concept annotations as its inputs *during training* (Figure 2 top). We also learn a composition model with predicted concept activations from a VL pretrained model (Figure 2 middle). We then compute the performance gap between the two models when the true primitive concept annotations are provided *during evaluation* (Figure 2 bottom): The gap is small when the learned composition correctly maps the true primitive concept annotations to their corresponding composite concepts, as can be achieved by a ground truth composition model. The composition concept mapping (CompMap) framework serves as a proxy benchmark to measure how well pretrained VL models learn primitive concepts. Our benchmark does not require exhaustive primitive concept labeling on the evaluation dataset, since the true composition mapping can be designed by experts for each composite concept (e.g. a *red banana* is *red* and a *banana*), or learned from a few training examples with true concept annotations.

We focus our study on three recent state-of-the-art VL models: (CLIP) (Radford et al., 2021), ViLT  (Kim et al., 2021), and ALBEF  (Li et al., 2021). They represent three categories of VL models: no cross-modal fusion (CLIP), early fusion (ViLT), and late fusion (ALBEF). To learn the composition models, we consider tasks where the target labels are composite concepts, such as in fine-grained visual recognition  (Wah et al., 2011) (where the primitives are color, shape of the object parts) and object state recognition (where the primitives are objects and their attributes). Our study reveals both limitations and promises of VL pretrained models. For limitations, we observe that the learned composition models are *not* interpretable, indicating that the model's conceptualization does not align well with how humans prefer to think about these composite concepts. For promises, we find that the primitive concept activations preserve the visual information useful for visual recognition tasks, such as few-shot fine-grained visual recognition (Tokmakov et al., 2019) and zero-shot compositional learning (Purushwalkam et al., 2019).

To summarize, we make the following contributions: (1) We propose the Compositional Concept Mapping (CompMap) framework to quantitatively measure how well VL pretrained models learn primitive concepts, via the composition model learning task; (2) We perform extensive quantitative and qualitative studies based on our proposed framework, using a range of recent VL pretrained models; (3) Our analysis shows that composition models learned from pretrained VL models are not *interpretable*, indicating that the existing VL models do *not* learn interpretable primitive concepts. The models are nonetheless useful for visual recognition tasks. Both findings highlight the importance to improve VL pretraining by learning interpretable primitive concepts. We will release the code and models used in the paper.

---

[3]We acknowledge that this may not be applied to every composite concept in general.

[4]In this work, we suppose that primitive concepts and composite concepts are relative terms, where each composite concept can be decomposed into a set of descriptive primitive concepts (i.e., attributes).

(a)     A learned ground truth composition model trained with ground truth concept activations. During inference time, the model takes in ground truth concept activations and predict the correct bird class.

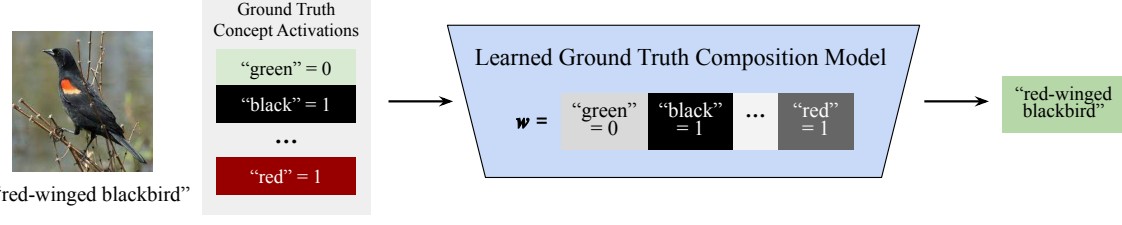

(b)     A composition model learned from concept activations predicted by a vision-and-language (VL) model. The model is affected by the green background, and thus learns a weight of 0.5 for concept "*green*".

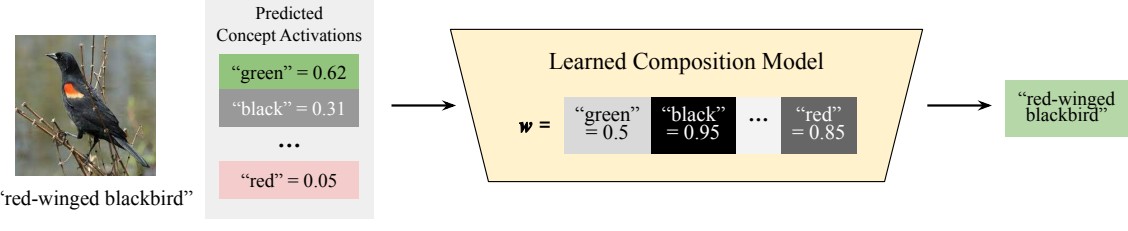

(c)     After training this learned composition model with predicted concept activations. We *intervene* this model with ground truth concept activations during inference time. The model predicts the wrong bird class because the learned weights reflect a positive correlation between the green background and *red-winged blackbird*.

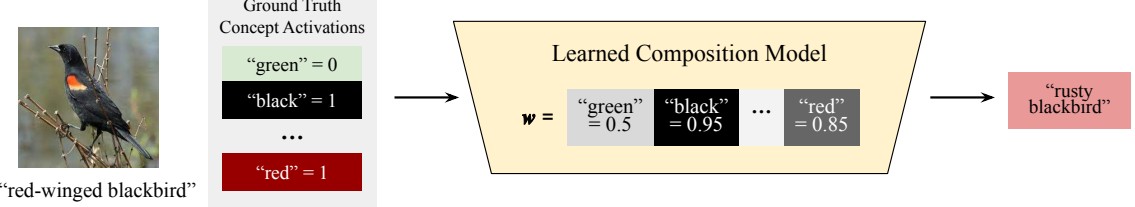

Figure 2: Illustration of a hypothetical scenario for composition learning, which aims to learn a linear mapping from primitive concepts (e.g. colors of a bird) into composite concepts (e.g. *a red-winged blackbird*). While the composition model can be reliably learned given true concepts (Figure a), the task can be challenging with noisy concept predictions. In this example, a pretrained VL model confuses the background color (*green*) with the wing color (*black* and *red*), resulting a composition model that happens to make a correct prediction with wrong reasons (Figure b). We quantify this phenomenon with "intervention", which feeds the true concept activations to a learned composition model for evaluation (Figure c), and measure the performance gap between the learned composition model and the ground truth composition model.

## 2   Related Work

**Vision and language pretraining.** The fast evolution of deep learning hardware enables researchers to train large deep neural networks on web-scale data (Devlin et al., 2018; Brown et al., 2020). One data source is vision and language pairs from the internet, such as images and captions (Sharma et al., 2018), or videos and speech transcripts (Miech et al., 2020; Radford et al., 2021). These vision-language datasets are usually not manually annotated. Representations learned by VL models can be transferred to a diverse range of tasks, such as image or video captioning (Lu et al., 2019; Sun et al., 2019) and visual question answering (Lu et al., 2019; Bommasani et al., 2021). These pretrained models can directly recognize composite concepts, such as human actions (Sun et al., 2019) or object categories (Radford et al., 2021) with text prompts. To our knowledge, we make the first attempt to study if pretrained VL models learn primitive concepts.

**Primitive concepts and their representation.** Based on the theory of concept that emphasizes symbolic representations, previous works found that neural networks satisfy multiple criteria of the theory, such as grounded behavior, but don't show a direct causal connection between the representations of constituent concepts and those of composite concepts. Visual and linguistic concepts are highly compositional and can be represented with primitive concepts. For human activities, the primitives are humans, objects, and their interactions (Ji et al., 2020). For objects, the primitives can be parts (Hoffman & Richards, 1984; Felzenszwalb et al., 2008) and their attributes (Parikh & Grauman, 2011; Kumar et al., 2009; Lampert et al., 2009; Lovering & Pavlick, 2022). For sentences, the primitives are often taken to be words and their grammatical relations (Socher et al., 2013).

To study the relations between the representations of composite and primitive concepts, measurements on compositionality have been proposed (Andreas, 2019; Keysers et al., 2019; Tokmakov et al., 2019). One common technique is to rely on representation arithmetic, where the representation of a composite concept is expected to be reconstructed by its primitives (e.g. summation (Andreas, 2019)). This measurement requires having a fixed set of primitive concepts and knowing the composition model from primitive to composite concepts, both of which might not hold for the composite concepts observed by a pretrained VL model. Instead, our paper assumes that the primitives can be selected *on-the-fly* based on different sets of composite concepts, and the composition model can be learned by a linear classifier.

**Using and learning primitive concepts.** Primitive concepts have wide applications in machine learning models, such as neural symbolic reasoning (Yi et al., 2018; Mao et al., 2019; Andreas et al., 2016b;a), few-shot (Tokmakov et al., 2019) or zero-shot (Parikh & Grauman, 2011; Lampert et al., 2009) visual recognition, building interpretable models (Kim et al., 2018; Ghorbani et al., 2019; Koh et al., 2020), or simply as a visual descriptor (Li et al., 2014). They require annotations of the primitives to build primitive classifiers before the primitive concepts can be incorporated into the overall framework. In this paper, we are interested in whether the primitive concepts can be recognized by a pretrained VL models via zero-shot prompts, a process that does not require any manual annotations, or task-specific *prompt tuning* (Yao et al., 2021).

## 3 Method

This section describes our proposed framework to measure quality of the learned primitive concept predicted by VL pretrained models. We first introduce the two stages of our proposed framework, concept prompting and composition learning. We name our proposed framework *Comp*ositional Concept *Map*ping, or *CompMap*. We then discuss our concept prompting setup and our two evaluation benchmarks. An illustration of the overall framework can be found in Figure 1.

### 3.1 Concept Prompting

A typical pretrained VL model consists of a text encoder $\mathcal{G}$ and an image encoder $\mathcal{F}$. Parts of the encoders can be shared to fuse vision-language signals. During VL pretraining, a score $s(x, t)$ is computed to measure the "compatibility" between an image $x$ and a sentence $t$, such as with cosine similarity between the encoded $\mathcal{F}(x)$ and $\mathcal{G}(t)$. The score should be high for the positive pairs of images and their corresponding descriptions and low for the negative pairs. One appealing property of vision-language pretrained models is that they provide a natural language-based interface to perform visual recognition tasks. This can be achieved via natural language *prompts*, by asking a VL model to measure how well a prompt, such as "a photo of a `[concept]`", is compatible with the content of an image. These natural language prompts are usually evaluated in zero-shot image classification tasks, where composite visual concepts, such as fine-grained object categories in ImageNet (Deng et al., 2009), are provided as candidate concepts to the VL model. Our paper follows the same concept prompting strategy, and assumes that if a VL model manages to learn primitive visual concepts, it can perform zero-shot primitive concept recognition via the corresponding natural language prompts. However, due to the lack of suitable benchmarks, it is challenging to directly evaluate the classification accuracy of primitive visual concepts.

We rely on two sources to obtain the vocabularies of primitive concepts: (1) Expert-defined primitive concepts, such as the color, shape, texture of an object, or the attributes of object parts. An example is

fine-grained bird species recognition; (2) Human-annotated composite concepts. For example, given a composite concept *ripe apple*, we use its attribute *ripe* and object category *apple* as the primitive concepts. When picking the composite concepts (e.g. fine-grained bird species, or objects and their attributes), we focus on the concepts which can be unambiguously described using a collection of non-overlapping primitive concepts. More formally, we denote a general visual concept as $c \in \mathcal{C}$, a primitive concept as $p \in \mathcal{C}_p$, and a composite concept as $q \in \mathcal{C}_q$. By definition we have $\mathcal{C}_p \subset \mathcal{C}$, $\mathcal{C}_q \subset \mathcal{C}$, and $\mathcal{C}_p \cap \mathcal{C}_q = \emptyset$. Prior work has worked on datasets with human-annotated images with primitive and composite concepts (Wah et al., 2011; Isola et al., 2015), while our goal is to validate if the primitive concepts can be "annotated" automatically with the VL pretrained models. For a primitive concept $p$, we fit it into a template sentence $t(p)$. A concept representation can be computed with the score function $e_p = s(x, t(p))$, as illustrated in Figure 1 (left). We can repeat the process for all primitive concepts to obtain the predicted primitive concept activations $\mathbf{e}^{\text{pred}} = [e_1, ..., e_N]$. In appendix, we study the impact of the choice of prompt template $t(\cdot)$.

## 3.2 Composition Learning

We denote the process of inferring composite concepts (e.g. fine-grained bird species) from primitive concepts (e.g. colors and shapes of a bird beak) as *composition*. In this work, we assume that the primitive concepts are composable, and there exists a true composition for any composite concept $q$, given an expressive set of candidate primitive concepts $\mathcal{C}_p$. As illustrated in Figure 2, a true composition model can be represented in the form of a linear classifier, where the primitive concepts required for the composition are assigned with $w_i = 1$ (or other positive weights), and the primitive concepts to be ignored are assigned with $w_j = 0$.

We are interested in *composition learning*: Given a set of paired primitive concept activations $\mathbf{e}$ along with composite concept labels, can we learn a composition model $\mathcal{H}$ that correctly maps the primitive concepts into the composite concepts, as achieved by the true composition model? Formally, we want to predict a composite concept $q$ with a linear classifier $\mathcal{H}_q(\mathbf{e}) = \mathbf{w}_q^\mathsf{T} \cdot \mathbf{e}$, where each element $e_i$ in $\mathbf{e}$ corresponds to how likely a concept $p_i$ is to be present, and can be given by an oracle as ground truth $\mathbf{e}^{\text{gt}}$, or predicted by a pretrained VL model $\mathbf{e}^{\text{pred}}$ (see Section 3.1). The classifier can be trained with a contrastive learning objective (Radford et al., 2021), which pushes the positive pairs of activated primitive concepts and the composition of their corresponding composite concept to be more similar ($\mathcal{H}_q$ to be higher), and the negative pairs to be less similar. Given a fixed set of mutually-exclusive composite concepts, this objective is the same as Softmax cross-entropy.

The composition model is regarded as a "proxy" to measure the quality of primitive concepts learned by VL models. We quantify the quality of the learned concepts with two metrics: *usefulness* and *interpretability*. First, the composition model can be directly evaluated for its *usefulness* as a classifier, i.e. the classification accuracy when evaluated on an unseen test set. Second, we want to measure whether the composition model is *interpretable*, meaning that the primitive concepts necessary to derive a composite concept are selected by the model and that the learned composition model is not "right for the wrong reason".

**Quantifying usefulness.** Intuitively, we claim the learned composition model to be useful if it can correctly map the predicted primitives $\mathbf{e}^{\text{pred}}$ into composite concepts. A useful composition model should sufficiently learn to compose these primitive concepts such that it would be able to derive unseen composite concepts correctly, even when deriving novel composite concepts. We use multiple evaluation metrics (e.g., classification accuracy) in composite concept recognition tasks to quantitatively measure the usefulness of a trained composition model; however this is not the main focus of the paper, but rather to validate that the linear classifier adopted by our composition framework does not come at the cost of performance.

**Quantifying interpretability.** We assume (and validate experimentally in Section 5.1) that when the ground truth primitive concept activations $\mathbf{e}^{\text{gt}}$ and their corresponding composite concept $q^{\text{gt}}$ are available, the composition from primitive concepts to composite concepts can be reliably learned by a linear composition model $\mathcal{H}_q^{\text{gt}}(\mathbf{e})$ for any composite concept $q$ (Figure 2(a)).

If pretrained VL models do learn primitive concepts well, their predicted primitives $\mathbf{e}^{\text{pred}}$ should learn a composition model $\mathcal{H}_q^{\text{pred}}$ *similar* to $\mathcal{H}_q^{\text{gt}}$, based only on pairs of $\mathbf{e}^{\text{pred}}$ and $q^{\text{gt}}$ (Figure 2(b)): Namely, $\mathcal{H}_q^{\text{pred}}$

should mostly pick the true primitive concepts necessary to compose $q$, just like $\mathcal{H}_q^{\text{gt}}$ does. We consider this "similarity" to the true composition as *interpretability* of a learned composition model.

To quantitatively measure the interpretability of a composition model $\mathcal{H}_q^{\text{pred}}$, we compare its performance with the corresponding ground truth composition $\mathcal{H}_q^{\text{gt}}$. Rather than forcing $\mathcal{H}_q^{\text{pred}}$ to behave exactly as the true composition for arbitrary inputs $\mathbf{e}$, we relax the constraint and only require it to behave similarly on true primitives $\mathbf{e}^{\text{gt}}$. Formally, we define the interpretability metric as: $\Delta = \texttt{Metric}\left(\mathcal{H}_q^{\text{gt}}(\mathbf{e}^{\text{gt}})\right) - \texttt{Metric}\left(\mathcal{H}_q^{\text{pred}}(\mathbf{e}^{\text{gt}})\right)$, where $\texttt{Metric}(\cdot)$ is the performance evaluation metric such as classification accuracy. We implicitly assumes that $\mathcal{H}_q^{\text{gt}}(\mathbf{e}^{\text{gt}})$ would perform nearly perfectly (e.g. accuracy of 100%). Alternatively, we can normalize the interpretability metric by the performance of $\mathcal{H}_q^{\text{gt}}(\mathbf{e}^{\text{gt}})$. Replacing the model inputs with ground truth concepts given by an oracle can be seen as a special case of *model intervention* (Koh et al., 2020) (Figure 2(c)).

**Discussion.** A possible alternative to quantitatively measure if VL pretrained models learn primitive concepts is to directly evaluate models' performance when predicting all primitive concepts. However, this evaluation requires exhaustive labeling of primitive concepts on a given image, which is often not available (e.g. an image of "red apple" in MIT-States (Dosovitskiy et al., 2021) can also be "ripe" or "round", which are not labeled.). Therefore, the composition model learning and the *usefulness* and *interpretability* metrics serve as proxy benchmarks to measure how well pretrained VL models learn primitive concepts: they offer two aggregated metrics by jointly considering the whole set of primitive concepts, as opposed to measuring individual "performance" on each primitive concepts.

### 3.3 Benchmarks

We consider two image classification benchmarks to learn the composition models, and measure model usefulness and interpretability. Our desired benchmarks should provide: (1) annotated image-level labels that are composite concepts; (2) vocabulary for the relevant primitive concepts.

**Compositional zero-shot learning (CZSL)** by Isola et al. (2015) aims to evaluate if a classifier trained to predict composite concepts generalizes to novel compositions, where each composite concept is a pair of object attribute (*red*) and category (*apple*). It provides a challenging setup for us to evaluate the *usefulness* and *interpretability* of a composition model.

Given the set of composite concept labels $\mathcal{C}^s$ during training and $\mathcal{C}^t$ during evaluation, CZSL considers the scenario when $\mathcal{C}^s \neq \mathcal{C}^t$, but both share the same set of object attributes and categories. We consider the generalized CZSL setup, where $\mathcal{C}^s$ is a true subset of $\mathcal{C}^t$. We also consider both the "closed-world" version and the "open-world" version of generalized CZSL. In the closed-world setting, $\mathcal{C}^t$ is constraint to be a relatively small set of composite concepts. In the open-world setting, $\mathcal{C}^t$ contains every combination of object attribute and category, and is therefore much more challenging due to the large output space.

**Fine-grained visual recognition**. We evaluate whether a true composition model could be learned in a few-shot or full-shot setting for fine-grained visual recognition tasks (Wah et al., 2011). Our intuition is that fine-grained concepts could potentially be defined by the details given by primitive concepts, such as shape or color of bird body parts. We adopt the standard few-shot learning setup, in which the composition model needs to discriminate $n$ composite concepts by training on only $k$ images from each composite concept.

## 4 Experimental Design

In this section, we introduce our datasets and metrics for the datasets (Section 4.1), and implementation details (Section 4.2). We elaborate how we collect concept activations with different model architectures (CLIP (Radford et al., 2021), ViLT (Kim et al., 2021) and ALBEF (Li et al., 2021)) in Appendix A.2.

### 4.1 Experimental Setup

**Datasets.** We conduct experiments on two image classification benchmarks where the target labels are composite concepts. The first dataset is MIT-States (Dosovitskiy et al., 2021), which contains 53K images of

115 attributes and 245 objects. Each image is labeled with (object, attribute) tuples. We use the standard splits from Purushwalkam et al. (2019). The training split has 30K images of 1262 seen attribute-object compositions, the validation split has 10K images of 300 seen and 300 unseen compositions, and the test split has 13K images of 400 seen and unseen compositions. We report results on the validation and test splits.

The other dataset is Caltech-UCSD Birds-200-2011 (CUB) (Wah et al., 2011), which contains 11788 photographs of 200 mainly North American bird species. We use the standard training/testing split from Hilliard et al. (2018). Each image is also annotated with attributes corresponding to 28 different categories, such as throat color, wing shape, etc. There are 312 possible binary attributes in total. We adopt the attribute denoising setup from Koh et al. (2020).

**Metrics.** For the MIT-States dataset, we follow the evaluation protocol from Purushwalkam et al. (2019) whose goal is to mitigate models' bias on seen compositions. The evaluation protocol has four metrics: (1) Unseen-Seen Area Under the Curve ($AUC$), (2) best accuracy on data samples of seen compositions (*best seen*), (3) best accuracy on data samples of unseen compositions (*best unseen*), and (4) best harmonic mean (*best HM*) of seen and unseen accuracy (2, 3). For the CUB dataset, we follow the standard practice in $n$-way $k$-shot evaluation (Snell et al., 2017) and report mean accuracy over the 600 sampled tasks run for each setup. In each task, the $n$ classes and $k$ examples are chosen at random. All metrics measure usefulness, where higher values correspond to being more "useful".

Since a composition is trained only on seen composite concepts, the seen concepts can evaluate much better or worse than the unseen ones on validation and test splits. We follow Purushwalkam et al. (2019) and apply calibration biases, which are scalars to be added to the predicted scores of unseen concepts (See Appendix A.6.2 for more details). Specifically, with a larger bias, the accuracy of unseen concepts tends to increase, while that of seen concepts tends to decrease. We vary the values of the calibration bias, and compute a list of accuracy scores of seen concepts and another list for unseen concepts. $AUC$ is the area under the curve of unseen accuracy and seen accuracy; *Best seen* and *best unseen* are the best accuracy in the lists of accuracy scores of seen and unseen concepts. *Best harmonic mean* is the highest harmonic mean computed by

$$HM = \sqrt{ACC_{seen} \cdot ACC_{unseen}} \tag{1}$$

The *interpretability* is then quantified as the gaps between ground truth composition model and a learned composition model for the above metrics.

## 4.2 Implementation Details

We choose three recently open-sourced VL models: CLIP (Radford et al., 2021), ViLT (Kim et al., 2021), and ALBEF (Li et al., 2021) for this study. They represent three categories of VL models: no cross-modal fusion (CLIP), early fusion (ViLT), and late fusion (ALBEF). For CLIP, we use the pretrained CLIP with a ViT-B/32 (Dosovitskiy et al., 2021) visual encoder and a transformer-based (Vaswani et al., 2017) text encoder. For ViLT, we use ViLT-B/32 pretrained with masked language modeling and image-text matching objectives. For ALBEF, we use ALBEF with a ViT-B/16 (Dosovitskiy et al., 2021) visual encoder and a BERT (Devlin et al., 2018) text encoder. This checkpoint is trained on Conceptual 12M dataset (Changpinyo et al., 2021). We follow the default setups to compute the concept activations.

For MIT-States, a contrastive objective is used to train the composition models. We apply two *linear* projection layers on the primitive concept activations and the text embeddings of target composite concepts respectively, to embed them into a shared space. This approach works with both closed-world and open-world settings of CZSL, and maintains the linear composition assumption. During training, a subset of composite concepts are supposed to be unknown, so we remove the unseen composite concepts (i.e. $|\mathcal{C}^s| = 1,262$ for training; $|\mathcal{C}^t| = 1,962$ for closed-world evaluation, and $|\mathcal{C}^t| = 28,175$ for open-world evaluation).

For CUB, a logistic regression model is used to train the composition models since the number of target composite concepts is fixed. We used the default sklearn hyperparameters and $L2$ regularization of 1 for all experiments. We observed that the performances are robust against the choice of logistic regression hyperparameters. Due to annotation inconsistencies, we follow Koh et al. (2020) and denoise the primitive

Table 1: Results of CZSL task on MIT-States in the *closed-world* setting. AUC (%) are computed using precision at $k$=1,2,3. Best seen and unseen accuracies, and best harmonic mean of the two are reported. Composition models are learned from ground truth primitive concepts $\mathbf{e}^{\text{gt}}$ (first row), or from VL model concept prompting $\mathbf{e}^{\text{pred}}$. Training and Evaluation columns specify model inputs during training and inference time respectively, where Prim stands for primitive concept activations, and GT stands for ground truth primitive concept activations. During inference time, when both Prim and GT are checked, it means ground truth primitives are activated, and all dimensions not activated by the ground truth primitives are left as the primitive concept activations. Usefulness is measured when the same predicted concepts $\mathbf{e}^{\text{pred}}$ is used for evaluation (rows in blue). Interpretability is measured as the performance gap from learned GT composition model (*Oracle-Prim*) when $\mathbf{e}^{\text{gt}}$ is provided as "intervention". (See Table 3 for a more detailed analysis of this performance gap.)

| Method | Training Prim | Training GT | Evaluation Prim | Evaluation GT | Val AUC $k$=1 | Val AUC 2 | Val AUC 3 | Test AUC 1 | Test AUC 2 | Test AUC 3 | Seen | Unseen | HM |
|---|---|---|---|---|---|---|---|---|---|---|---|---|---|
| Oracle-Prim | | ✓ | | ✓ | 99.9 | 99.7 | 99.7 | 99.9 | 99.9 | 99.9 | 100.0 | 100.0 | 99.9 |
| CLIP-Prim | ✓ | | ✓ | | 8.2 | 17.0 | 24.1 | 6.9 | 15.6 | 22.8 | 34.0 | 27.9 | 20.4 |
| CLIP-Interv(Full) | ✓ | | | ✓ | 32.2 | 49.1 | 58.0 | 30.0 | 49.3 | 59.6 | 52.3 | 66.0 | 47.7 |
| CLIP-Interv(Partial) | ✓ | | ✓ | ✓ | 35.8 | 53.4 | 62.9 | 32.6 | 53.7 | 63.2 | 54.1 | 68.2 | 49.4 |
| ViLT-Prim | ✓ | | ✓ | | 4.7 | 11.1 | 16.9 | 3.8 | 9.7 | 15.2 | 25.4 | 20.8 | 15.1 |
| ViLT-Interv(Full) | ✓ | | | ✓ | 8.6 | 17.1 | 23.6 | 4.1 | 10.2 | 14.4 | 21.7 | 24.8 | 16.3 |
| ViLT-Interv(Partial) | ✓ | | ✓ | ✓ | 20.1 | 33.8 | 43.1 | 12.9 | 25.5 | 35.1 | 39.0 | 41.8 | 28.7 |
| ALBEF-Prim | ✓ | | ✓ | | 7.1 | 15.6 | 23.1 | 5.8 | 13.9 | 21.2 | 32.9 | 25.1 | 18.5 |
| ALBEF-Interv(Full) | ✓ | | | ✓ | 36.8 | 52.7 | 63.6 | 37.9 | 57.2 | 64.8 | 58.5 | 71.6 | 53.0 |
| ALBEF-Interv(Partial) | ✓ | | ✓ | ✓ | 38.7 | 56.5 | 68.3 | 41.1 | 61.8 | 70.6 | 61.3 | 73.0 | 55.8 |

concept annotations via majority voting. Thus, we use class-level concepts, which means that the bird attributes of the same bird species are always the same, regardless of the images of birds.

We design two sets of prompts to extract attribute/object/pair activations from the VL models. For MIT States, e.g., *"this is {ripe}"*, *"this is {apple}"*, and *"this is {ripe apple}"* are used to produce the corresponding activations. For CUB, we compute attribute and class activations by *"a photo of bird whose {bill shape} is {needle}"* and *"the bird is {bohemian waxwing}"*. These two CUB prompts have different templates since we observe some bird attribute categories can only be understood with the context of birds (e.g. size and shape of birds), and thus we include the language prior "bird" in both prompts. We explore more prompt templates in Appendix A.4 and find our observations to be robust.

## 5 Results

### 5.1 Interpretability of Learned Composition Models

We first explore if a composition model learned from concept activations is interpretable. We approach this problem by inspecting if the linear classifier learns a true composition for the composite concepts or it solves the task with some spurious correlations between the activations and the composite concepts. The true composition is learned from ground truth primitives, while a learned composition is learned from concept activations predicted by VL models.

We intervene on the learned compositions in two ways. The first method is intervention with ground truth primitives (*Interv(Full)*), which replaces the primitive concept activations with the binary ground truth primitives during inference. The comparison between *Interv(Full)* and a corresponding ground truth composition can reflect how feasible it is to approximate a ground truth composition from primitive concept activations. The second is intervention only on attributes activated by the ground truth primitives (*Interv(Partial)*). This method is similar to *Interv(Full)*, but all dimensions not activated by the ground truth labels are left as

Table 2: Results of $n$-way $k$-shot evaluation on CUB. Mean per-sample accuracy scores are reported. Composition models are learned from ground truth primitive concepts $\mathbf{e}^{\mathrm{gt}}$ (first row), or from VL model concept prompting $\mathbf{e}^{\mathrm{pred}}$. Training and Evaluation columns specify model inputs during training and inference time respectively, where Prim stands for primitive concept activations, and GT stands for ground truth primitive concept activations. During inference time, when both Prim and GT are checked, it means ground truth primitives are activated, and all dimensions not activated by the ground truth primitives are left as the primitive concept activations. Full Shot column refers to a 200-way full-shot evaluation that considers all the available data in CUB. Usefulness is measured when the same predicted concepts $\mathbf{e}^{\mathrm{pred}}$ is used for evaluation (rows in blue). Interpretability is measured as the performance gap from learned GT composition model (*Oracle-Prim*) when $\mathbf{e}^{\mathrm{gt}}$ is provided as "intervention".

| Method | Training Prim | Training GT | Evaluation Prim | Evaluation GT | $n=5$ $k$=1 | 5 | $n=10$ 1 | 5 | $n=100$ 1 | 5 | $n=200$ 1 | 5 | Full Shot |
|---|---|---|---|---|---|---|---|---|---|---|---|---|---|
| Oracle-Prim | | ✓ | | ✓ | 99.8 | 100.0 | 99.9 | 99.9 | 98.9 | 98.9 | 98.0 | 97.9 | 98.0 |
| CLIP-Prim | ✓ | | ✓ | | 76.5 | 87.3 | 60.9 | 78.0 | 21.0 | 43.0 | 13.6 | 33.2 | 52.6 |
| CLIP-Interv(Full) | ✓ | | | ✓ | 61.6 | 66.0 | 43.1 | 50.8 | 9.4 | 13.4 | 5.1 | 7.8 | 8.0 |
| CLIP-Interv(Partial) | ✓ | | ✓ | ✓ | 75.4 | 85.4 | 59.4 | 75.2 | 17.5 | 32.9 | 10.9 | 23.4 | 32.9 |
| ViLT-Prim | ✓ | | ✓ | | 70.0 | 83.4 | 54.0 | 71.3 | 14.8 | 28.8 | 8.9 | 19.7 | 40.1 |
| ViLT-Interv(Full) | ✓ | | | ✓ | 54.3 | 60.7 | 37.7 | 45.7 | 6.5 | 8.6 | 3.5 | 4.7 | 9.2 |
| ViLT-Interv(Partial) | ✓ | | ✓ | ✓ | 64.9 | 75.8 | 48.0 | 60.0 | 9.6 | 15.6 | 5.3 | 9.4 | 19.3 |
| ALBEF-Prim | ✓ | | ✓ | | 73.2 | 85.2 | 58.3 | 74.4 | 17.0 | 36.3 | 10.7 | 27.2 | 44.6 |
| ALBEF-Interv(Full) | ✓ | | | ✓ | 57.0 | 63.9 | 39.3 | 46.5 | 7.0 | 9.5 | 3.7 | 5.2 | 7.6 |
| ALBEF-Interv(Partial) | ✓ | | ✓ | ✓ | 69.9 | 80.0 | 52.4 | 66.3 | 12.0 | 24.6 | 7.0 | 16.9 | 30.2 |

Table 3: Measuring interpretability $\Delta$ on MIT-States (Left) and CUB (Right). $\Delta$ is the performance gap between learned GT composition model (*Oracle-Prim*) and Interv(Full). On CUB, we show the $\Delta$ for $n$-way 5-shot evaluation. The lower $\Delta$ is, the more interpretable the composition model is.

| Model | AUC | $\Delta$ Seen | Unseen | HM |
|---|---|---|---|---|
| CLIP | 69.9 | 47.7 | 34.0 | 52.2 |
| ViLT | 95.8 | 78.3 | 75.2 | 83.6 |
| ALBEF | 62.0 | 41.5 | 28.4 | 46.9 |

| Model $n \rightarrow$ | $\Delta$ 5 | 100 | 200 | FS |
|---|---|---|---|---|
| CLIP | 34.0 | 85.5 | 90.1 | 90.0 |
| ViLT | 39.3 | 90.3 | 93.2 | 88.8 |
| ALBEF | 36.1 | 89.4 | 92.7 | 90.4 |

the activations predicted by the VL model. The comparison between *Interv(Partial)* and *Interv(Full)* shows the spurious correlations learned between non-activated primitives and composite concepts.

Table 1 and Table 2 summarize the results on MIT-States and CUB. First of all, the composition models learned from ground truth primitive concept activations $\mathbf{e}^{\mathrm{gt}}$ and their corresponding composite concept $q_{\mathrm{gt}}$ perform almost perfectly (*Oracle-Prim*). This validates our hypothesis that a true composition can be learned from $\mathbf{e}^{\mathrm{gt}}$ and $q_{\mathrm{gt}}$. We also observe two consistent patterns on both CZSL and few-shot learning tasks. First, with *Interv(Full)* intervention, all models perform worse than the ground truth composition (Table 3), which generally reaches perfect performance. This implies that the predicted primitive concept activations and the composition models learned from them are not interpretable. Second, by comparing *Interv(Partial)* intervention with *Interv(Full)*, all models increase in performance, indicating that the models utilize activations from the irrelevant primitive concepts to predict the composite concepts.

When we compare *Prim* with *Interv(Full)*, we see that *Interv(Full)* performs better than *Prim* in MIT-States, but worse than *Prim* in CUB. This indicates that the learned compositions in MIT-States are able to identify positive correlations between the composite concepts and their primitives, while the learned compositions in CUB are not able to identify such positive correlations sufficiently. We attribute this difference to the

Table 4: Quantitative analysis of the learned weights of ground truth composition learned from binary ground truth primitive concepts and the learned weights of compositions learned from concept activations predicted by CLIP. Accuracy measures whether the most activated weights align with the ground truth primitive concepts. If there are $k$ ($k > 1$) ground truth primitive concepts, we consider the top $k$ most activated weights. Instance-level accuracy (Acc-Instance)considers each alignment between the most activated weights with the ground truth weigths as individual instance. Class-level accuracy (Acc-Class) requires a composition model to correlate all the primitive concepts of a composite concept correctly. Acc-Attribute/-Object corresponds to the instance-level accuracy over attributes and objects in MIT-States. There are significant gaps between the learned ground truth composition models and composition models learned from CLIP.

| Method | MIT-States | | | | CUB | |
|---|---|---|---|---|---|---|
| | Acc-Instance | Acc-Attribute | Acc-Object | Acc-Class | Acc-Instance | Acc-Class |
| Oracle-Prim | 99.6 | 99.5 | 99.8 | 99.3 | 96.5 | 97.0 |
| CLIP-Prim | 51.4 | 34.3 | 68.5 | 22.5 | 37.6 | 0.0 |

gap of the complexity in these two tasks, as a composition model for MIT-States only needs to capture two primitives (an attribute and an object) for each composite concept, while a composition model for CUB must capture multiple primitives for each bird class.

## 5.2 Analyses of Composition Models

We now directly analyze the weights of the learned composition models. We consider an accuracy metric, which measures the percentage of the primitive concepts with the highest learned weights actually belong to the true composition model. The results are shown in Table 4. The high accuracy scores (e.g., class-level accuracy of 99.3% for MIT-States) for *Oracle-Prim* show that these models can learn the correlation between the primitive concepts and the composite concepts almost perfectly. We observe that there is significant gap between *Oracle-Prim* and the composition models learned from CLIP-predicted primitive concept activations (*CLIP-Prim*), indicating that the learned *CLIP-Prim* models' behaviors are not similar to the true composition model (see Figure 3). We also notice that *CLIP-Prim* reaches class-level accuracy of 0% on CUB (versus 97.0% for *Oracle-Prim*. We attribute this to the complexity of CUB, since a composition model needs to correlate all 28 bird attribute categories correctly for each bird class, which is significantly more difficult than MIT-States (only 2 primitive concepts – attribute and object). This may also explain the performance degradation of *CLIP-Interv(Full)* from *CLIP-Prim* on CUB (Table 2).

In order to understand if extracting the primitive concepts related to small regions is the bottleneck, we carry out a quantitative analysis on CUB dataset to study the relationship between the region size of the concepts and the correlation between primitives and composite concepts. The correlation is quantified by accuracy metric, which measures whether the most activated weights align with the ground truth primitive concepts. We observe no strong correlation between the region size of the primitive concepts and how well *CLIP-Prim* can correlate them to the composite concepts (Figure 4). For instance, even though bird wings are larger than their napes, the accuracy on nape color is significantly higher than on wing color (56.4% versus 41.7%).

## 5.3 Usefulness of Composition Models

We look into the classification performance based on the primitive concept activations. The primitive concepts are deemed useful if the composition learned on top of them achieves good classification performance.

Table 5 shows the results on MIT-States in closed-world setting. We observe that the zero-shot CLIP already achieves on par or better performance than the previous approaches, which confirms the observation that CLIP learns composite concepts. When we use a pretrained VL model with our CompMap framework, the performance of the composition models perform competitively against previous state-of-the-art methods (i.e., *CLIP-Prim* has a test AUC of 6.9), even though our CompMap has a very simple architecture. With

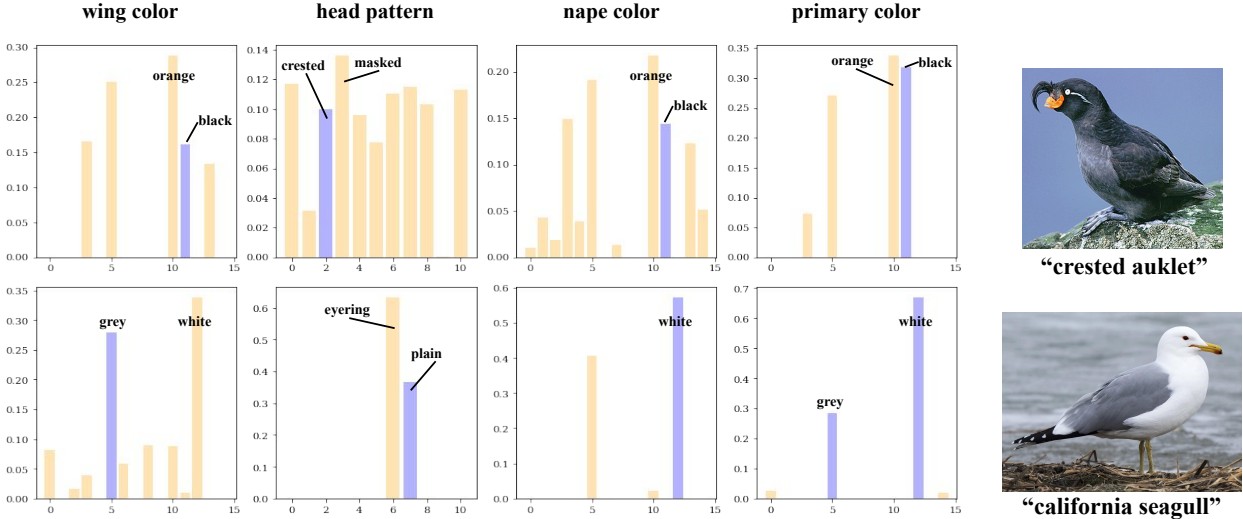

Figure 3: Visualization of the learned weights of composition model learned from CLIP-predicted concept activations (*CLIP-Prim*) on CUB dataset. X-axes stands for primitive concepts. Y-axes shows the likelihood for a primitive concept to be correlated with a composite concept. In each subplot, the blue bars highlight the ground truth primitive concepts. We observe that *CLIP-Prim* models are not similar to the true composition model, which can correlate primitive concepts for each composite concept correctly.

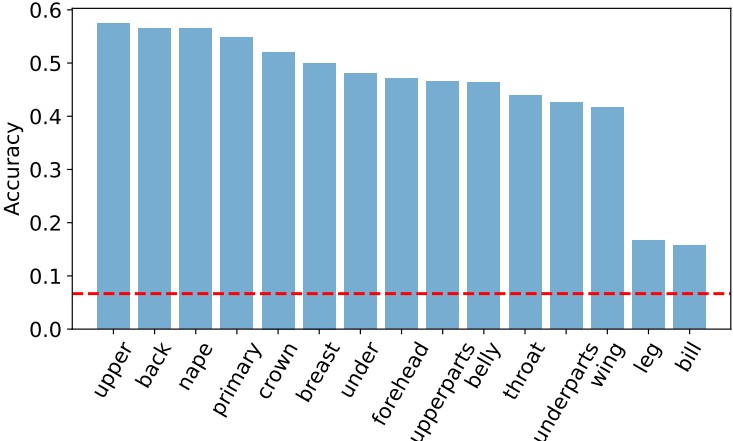

Figure 4: Analysis of the correlation between region size of primitive concepts and how well a learned composition model (*CLIP-Prim*) can match the primitives to the composite concepts on CUB dataset. This analysis focuses on color-related concepts. X-axis stands for primitive concepts. Y-axis shows the accuracy metric, which measures whether the most activated weights align with the ground truth primitive concepts. Red dotted line is the random baseline. We observe no strong correlation between region size of primitive concepts and how well *CLIP-Prim* can match the primitives to the composite concepts.

Table 5: Results of generalized CZSL on MIT-States in the closed-world setting. We observe that CLIP with CompMap performs the best among the selected VL models, and it outperforms previous methods.

| Method          Top $k \rightarrow$ | Val AUC | | | Test AUC | | | Seen | Unseen | HM |
|---|---|---|---|---|---|---|---|---|---|
|  | 1 | 2 | 3 | 1 | 2 | 3 |  |  |  |
| AOP (Nagarajan & Grauman, 2018) | 2.5 | 6.2 | 10.1 | 1.6 | 4.7 | 7.6 | 14.3 | 17.4 | 9.9 |
| LE+ (Nagarajan & Grauman, 2018) | 3.0 | 7.6 | 12.2 | 2.0 | 5.6 | 9.4 | 15.0 | 20.1 | 10.7 |
| TMN (Purushwalkam et al., 2019) | 3.5 | 8.1 | 12.4 | 2.9 | 7.1 | 11.5 | 20.2 | 20.1 | 13.0 |
| SymNet (Li et al., 2020) | 4.3 | 9.8 | 14.8 | 3.0 | 7.6 | 12.3 | 24.4 | 25.2 | 16.1 |
| CompCos (Mancini et al., 2021) | / | / | / | 4.5 | / | / | 25.3 | 24.6 | 16.4 |
| CGE$_{ff}$ (Naeem et al., 2021) | 6.8 | / | / | 5.1 | / | / | 28.7 | 25.3 | 17.2 |
| CLIP (zero-shot) | 5.8 | 13.0 | 19.1 | 5.5 | 12.7 | 18.8 | 25.0 | 30.5 | 18.3 |
| CLIP-Prim | 8.2 | 17.0 | 24.1 | 6.9 | 15.6 | 22.8 | 34.0 | 27.9 | 20.4 |
| ViLT-Prim | 4.7 | 11.1 | 16.9 | 3.8 | 9.7 | 15.2 | 25.4 | 20.8 | 15.1 |
| ALBEF-Prim | 7.1 | 15.6 | 23.1 | 5.8 | 13.9 | 21.2 | 32.9 | 25.1 | 18.5 |

CompMap framework, we also observe that CLIP outperforms ALBEF, and ALBEF outperforms ViLT, which matches to the order of their pretraining data size.

Table 6: Ablation study to investigate whether the usefulness of concept activations comes from the primitive concept prompting, or the powerful CLIP visual encoder. We compare CLIP-Prim, which applies a projection matrix generated based on the text prompt embeddings, with the direct application of CLIP image embedding, and a learned or random projection of the CLIP image embedding.

| Method                              Top $k \rightarrow$ | Val AUC | | | Test AUC | | | Seen | Unseen | HM |
|---|---|---|---|---|---|---|---|---|---|
|  | 1 | 2 | 3 | 1 | 2 | 3 |  |  |  |
| CLIP-Prim | 8.2 | 17.0 | 24.1 | 6.9 | 15.6 | 22.8 | 34.0 | 27.9 | 20.4 |
| CLIP Image Embed. | 8.6 | 17.4 | 26.7 | 7.0 | 15.8 | 23.4 | 36.1 | 26.4 | 20.9 |
| CLIP Image Embed. + Learned Projection | 7.9 | 16.4 | 26.0 | 6.2 | 14.8 | 21.6 | 34.5 | 24.9 | 19.3 |
| CLIP Image Embed. + Random Projection | 7.5 | 16.1 | 23.0 | 6.2 | 14.2 | 21.2 | 34.5 | 25.1 | 19.3 |

Table A1 shows the results on MIT-States in open-world setting. We mainly focus on CLIP due to its superior performance in the closed-world setting. With a linear composition model trained on top of concept activations predicted by CLIP, all learned compositions outperform the previous state-of-the-art approach. However, most improvements come from better learning the seen concepts in the training split but not necessarily from better generalizability to the unseen concepts, implying that the learned compositions are still distracted by the infeasible attribute-object pairs.

Table A2 shows the results on CUB in 5-way $k$-shot learning setting. We observe that the learned composition with CLIP again performs the best among the selected VL models. Given that VL models with CompMap have simple model architecture, they still perform competitively against the previous state-of-the-art in the 1-shot and 5-shot setting, implying that the predicted concept activations by pretrained VL models are useful for such few-shot learning problems.

Finally, we conduct an ablation study to investigate if the usefulness comes from the application of concept prompting, or comes from the power of raw CLIP image embeddings. We consider three scenarios: First, we directly use the CLIP image embedding with dimension of 512, which has higher dimensions than the primitive concept activations (i.e., dimension of 360 for MIT-States). Since concept prompting can be viewed as applying a projection matrix on the CLIP image embedding, we also consider two alternatives for the projection matrix used by CLIP-Prim, which is generated based on text prompts: an end-to-end learned

projection, and a frozen random projection. In Table 6, we observe that *CLIP-Prim* only moderately outperforms the learned and random projection baselines, indicating that the usefulness mainly comes from the CLIP visual encoder, and not from the primitive concept prompting. The results are in line with our earlier observation that the learned composition models are not interpretable according to our metrics.

## 6 Conclusions

In this paper, we have proposed a framework for measuring how well vision-language pretrained models can learn primitive concepts, in terms of usefulness and interpretability. By conducting extensive experiments with recent VL pretrained models - CLIP, ViLT, and ALBEF - we observe that these models are able to learn primitive concepts that are useful for visual recognition tasks, but the learned composition models from primitive concepts to composite concepts are often not interpretable. Our study suggests that despite the strong performance of VL pretrained models for visual recognition tasks, more research is needed to improve their ability to better capture primitive concepts. We anticipate our framework serve as a meaningful step towards understanding the working mechanisms (and potentially biases) of VL pretrained models.

**Limitations.** Our interpretability metric measures the unnormalized performance gap between the composition models learned on ground truth primitive concepts (Oracle-Prim) and concept activations generated by pre-trained VL models. It implicitly assumes that Oracle-Prim achieves near perfect performance when predicting the composite concepts. When the assumption is not true, a solution is to normalize the interpretability metric by the performance of Oracle-Prim. Our work also makes the assumption that the concept activations predicted by pretrained VL models are well-calibrated. However, this may not always hold true (example shown in Appendix A.1). A possible solution for this mis-calibration issue is to post-process the predicted concept activations. We leave this as a future work.

## 7 Acknowledgements

We would like to thank all reviewers and the action editor for their valuable feedback. We would like to thank Calvin Luo and Lishuo Pan for their discussions and insights. This work is in part supported by Adobe, Meta AI, Samsung, and Richard B. Salomon Faculty Research Awards for C.S.

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

# A  Appendix

## A.1  Mis-calibration Example

We would like to thank the reviewers for this mis-calibration issue and example. When there are two VL models: **A** and **B**. Model **A** predicts well-calibrated and accurate values (e.g., for red birds, it predicts {red: 1, black: 0}). Model **B** predicts relatively smaller values for the correct concepts (e.g., for red birds, it predicts {red: 0.8, black: 0.2}; for black birds, it predicts {red: 0.2, black: 0.8}). We train composition models using the predicted concept activations from models **A** and **B**. For example, their composition models both learn a set of weights {red: 1, black: 0} for red birds, and correspondingly for black birds. Therefore, both composition models reach the same interpretability scores during the intervention. However, based on the predicted concept activations, model **A** understands primitive concepts better than model **B** does. One possible solution for this mis-calibration issue is to post-process the predicted concept activations.

## A.2  Concept Prompting with CLIP, ViLT and ALBEF

CLIP (Radford et al., 2021) is a vision-and-language (VL) pretrained models without cross-modal fusion layers. CLIP learns a textual encoder and an image encoder via the contrastive learning objective. It does not need an explicit architecture component to fuse visual and language signals.

ViLT (Kim et al., 2021) is a single-stream early-fusion VL pretrained model that simplies the visual embedding pipeline. The model takes a concatenated sequence of a pair of text sequence and image sequence (of patches) as input, and is trained with image-text matching (ITM), masked language modeling (MLM), and word patch alignment objectives.

ALBEF (Li et al., 2021) is a dual-stream late-fusion VL pretrained model that learns a text encoder, an image encoder, and a multimodal encoder. The text input and image input are passed into their corresponding encoders, and the multimodal encoder has a cross-attention layer that fuses the encoded text input and encoded image input together. ALBEF is trained with image-text contrastive learning (ITC), image-text matching (ITM), and masked language modeling (MLM) objectives.

Given a set of primitive concepts $\{p_1, ..., p_N\}$ and an image $x$, we obtain a concept activation $\mathbf{e}^{\text{pred}}$ of image $x$ with each of CLIP, ViLT, and ALBEF. For CLIP, we collect the text embedding of the prompted text of a primitive concept $p_i$, and the image embedding of image $x$. We treat the dot product between the image embedding and text embedding as $e_i$. We then repeat the process for all primitive concepts to obtain $\mathbf{e}^{\text{pred}} = [e_1, ..., e_N]$. For ViLT and ALBEF, we extract the ITM score for each concept $p_i$ and treat each ITM score as $e_i$. We then repeat the process for all primitive concepts to obtain $\mathbf{e}^{\text{pred}} = [e_1, ..., e_N]$. We notice that computing concept activations for ViLT is significantly time-consuming due to its single-stream model input format - a concatenated sequence of text and image inputs. We cannot precompute any text or image features, and thus need to pass the concatenated sequences for each single ITM score. For dual-stream models (CLIP and ALBEF), we directly cache the encoded images and encoded primitive concepts, and then compute the concept activations based on these precomputed features.

## A.3  Usefulness of Composition Models

Table A1 shows the results of generalized CZSL on MIT-States in the open-world setting. Table A2 shows the results of 5-way $k$-shot learning on CUB. Results on both tables demonstrate that the learned primitive concept activations are useful for the image classification tasks. More details are discussed in Section 5.3.

## A.4  Ablation Study of Prompts

We carry out an ablation study to explore the impact of prompts. Table A3 shows the prompts collected from CLIP. This set of 7 prompt templates is the best performing subset on ImageNet Deng et al. (2009) over 80 prompt templates.

Table A1: Results of generalized CZSL on MIT-States in the open-world setting for CLIP with CompMap. We observe that CLIP with CompMap outperforms previous methods on AUC , but it does not necessarily generalize well to unseen composite concepts.

| Method      Top $k \rightarrow$ | Val AUC | | | Test AUC | | | Seen | Unseen | HM |
|---|---|---|---|---|---|---|---|---|---|
| | 1 | 2 | 3 | 1 | 2 | 3 | | | |
| CompCos (Mancini et al., 2021) | / | / | / | 1.6 | / | / | 25.4 | 10.0 | 8.9 |
| CLIP-Prim | 2.6 | 6.0 | 9.1 | 2.1 | 5.2 | 8.0 | 33.7 | 9.9 | 10.8 |

Table A2: Results of 5-way $k$-shot learning on CUB. Mean accuracy across 600 sampled tasks for CompMap-All are reported for CLIP, ViLT, and ALBEF. CompMap provides competitive performance.

| Method      $n = 5$ | 1-shot | 5-shot |
|---|---|---|
| FEAT  (Ye et al., 2020) | 73.3 | 85.8 |
| DeepEMD  (Zhang et al., 2020) | 75.7 | 88.7 |
| RENet  (Kang et al., 2021) | 79.5 | 91.1 |
| S2M2_R  (Mangla et al., 2020) | 80.7 | 90.9 |
| $PEM_b$-NCM  (Hu et al., 2021) | 80.8 | 91.5 |
| CLIP-Prim | 74.1 | 87.5 |
| ViLT-Prim | 69.9 | 84.0 |
| ALBEF-Prim | 74.1 | 86.3 |

### A.4.1  Different Prompt Templates

We first explore how the *usefulness* and *interpretability* of learned composition models vary with different prompt templates. We perform concept prompting based on each prompt in Table A3. We then train $(7 \times 2 =)$ 14 composition models in total. We use these models to investigate how robust our observations in the main paper are.

Tables A4 and A5 show the results for *usefulness* as measured on MIT-States in closed-world and open-world settings respectively. We observe that the aggregated results have slightly higher mean values than what were reported in the main paper, and the standard deviations are small. This confirms that our observations are robust and the choice of prompts does not matter much for usefulness of the learned compositions. The gains over the best performing baselines are significant considering the low standard deviations in both settings.

Table A6 shows the results with or without intervention, a protocol we use to measure *interpretability* of the learned compositions. We observe that the results with intervention generally exhibit higher variances, indicating that the choice of prompts matters more for the interpretability of the learned compositions. However, the aggregated results are still far from those of the ground truth composition.

### A.4.2  Using More Prompts

We further examine the impact of using more prompts. Following what CLIP does to combine prompts, we combine text embeddings from multiple prompts together. We vary the number of prompts to be 1, 4, or 7, and train their corresponding composition models. In Table A7, we observe a positive correlation between the number of prompts used to generate the concept activations and the interpretability of the learned compositions. However, the impact on usefulness is marginal.

Table A3: All seven prompt templates we use.

| Prompt ID | Prompt |
|---|---|
| 1 | itap of a {} |
| 2 | a bad photo of the {} |
| 3 | a origami {} |
| 4 | a photo of the large {} |
| 5 | a {} in a video game |
| 6 | art of the {} |
| 7 | a photo of the small {} |

Table A4: Aggregated results (mean and standard deviation) over composition models trained on seven prompt templates on MIT-States in closed-world setting. Template source of single template means we use the same prompt template as the one used in the main paper.

| Template Source | Top $k \rightarrow$ | Val AUC | | | Test AUC | | | Seen | Unseen | HM |
|---|---|---|---|---|---|---|---|---|---|---|
| | | 1 | 2 | 3 | 1 | 2 | 3 | | | |
| | CGE$_{ff}$ | 6.8 | / | / | 5.1 | / | / | 28.7 | 25.3 | 17.2 |
| Single | CLIP-Prim | 8.2 | 17.0 | 24.1 | 6.9 | 15.6 | 22.8 | 34.0 | 27.9 | 20.4 |
| Aggregated | CLIP-Prim | $8.7 \pm 0.2$ | $18.1 \pm 0.3$ | $25.5 \pm 0.2$ | $7.1 \pm 0.2$ | $16.2 \pm 0.1$ | $23.7 \pm 0.2$ | $35.1 \pm 0.5$ | $27.7 \pm 0.3$ | $20.7 \pm 0.4$ |

Table A5: Aggregated results (mean and standard deviation) over composition models trained on seven prompt templates on MIT-States in open-world setting. Template source of single template means we use the same prompt template as the one used in the main paper.

| Template Source | Top $k \rightarrow$ | Val AUC | | | Test AUC | | | Seen | Unseen | HM |
|---|---|---|---|---|---|---|---|---|---|---|
| | | 1 | 2 | 3 | 1 | 2 | 3 | | | |
| | CompCos | - | - | - | 1.6 | - | - | 25.4 | 10.0 | 8.9 |
| Single | CLIP-Prim | 2.6 | 6.0 | 9.1 | 2.1 | 5.2 | 8.0 | 33.7 | 9.9 | 10.8 |
| Aggregated | CLIP-Prim | $2.9 \pm 0.1$ | $6.5 \pm 0.2$ | $9.7 \pm 0.2$ | $2.2 \pm 0.1$ | $5.3 \pm 0.2$ | $8.0 \pm 0.2$ | $34.0 \pm 0.7$ | $10.0 \pm 0.3$ | $11.3 \pm 0.2$ |

Table A6: Aggregated results (mean and standard deviation) over composition models trained on seven prompt templates on MIT-States in closed-world setting. We perform intervention to measure interpretability

| Template Source | Top $k \rightarrow$ | Val AUC | | | Test AUC | | | Seen | Unseen | HM |
|---|---|---|---|---|---|---|---|---|---|---|
| | | 1 | 2 | 3 | 1 | 2 | 3 | | | |
| | Oracle-Prim | 99.9 | 99.7 | 99.7 | 99.9 | 99.9 | 99.9 | 100 | 100 | 99.9 |
| Single (main paper) | CLIP-Prim | 8.2 | 17.0 | 24.1 | 6.9 | 15.6 | 22.8 | 34.0 | 27.9 | 20.4 |
| | CLIP-Interv(Full) | 32.2 | 49.1 | 58.0 | 30.0 | 49.3 | 59.6 | 52.3 | 66.0 | 47.7 |
| | CLIP-Interv(Partial) | 35.8 | 53.4 | 62.9 | 32.6 | 53.7 | 63.2 | 54.1 | 68.2 | 49.4 |
| Aggregated | CLIP-Prim | $8.7 \pm 0.2$ | $18.1 \pm 0.3$ | $25.5 \pm 0.2$ | $7.1 \pm 0.2$ | $16.2 \pm 0.1$ | $23.7 \pm 0.2$ | $35.1 \pm 0.5$ | $27.7 \pm 0.3$ | $20.7 \pm 0.4$ |
| | CLIP-Interv(Full) | $39.3 \pm 3.9$ | $58.6 \pm 3.5$ | $68.6 \pm 3.4$ | $39.9 \pm 3.0$ | $58.1 \pm 3.1$ | $68.4 \pm 2.6$ | $59.2 \pm 2.4$ | $72.7 \pm 2.4$ | $55.7 \pm 2.6$ |
| | CLIP-Interv(Partial) | $43.6 \pm 4.3$ | $64.0 \pm 4.3$ | $73.3 \pm 4.2$ | $44.0 \pm 3.4$ | $63.7 \pm 3.5$ | $73.3 \pm 3.1$ | $62.8 \pm 3.2$ | $75.2 \pm 2.1$ | $59.0 \pm 2.9$ |

## A.5 Ablation Study of Model Size

We carry out an ablation study to explore the impact of scaling law on how well the VL pretrained model (CLIP) can capture primitive concepts on MIT-States. The results are shown in Table A8. We observe that larger models can predict more useful concept activations. However, larger models do not improve the

Table A7: The impact of combining activations from multiple prompt templates on MIT-States in closed-world setting

| Template Source | Top $k \rightarrow$ | Val AUC | | | Test AUC | | | Seen | Unseen | HM |
|---|---|---|---|---|---|---|---|---|---|---|
| | | 1 | 2 | 3 | 1 | 2 | 3 | | | |
| | GT Composition | 99.9 | 99.7 | 99.7 | 99.9 | 99.9 | 99.9 | 100 | 100 | 99.9 |
| CLIP w/ 1 prompt | CLIP-Prim | 8.2 | 17.4 | 24.2 | 7.1 | 16.3 | 24.0 | 35.0 | 28.2 | 20.8 |
| | CLIP-Interv(Full) | 43.6 | 63.1 | 73.0 | 40.9 | 59.8 | 71.0 | 59.0 | 75.1 | 56.3 |
| | CLIP-Interv(Partial) | 47.2 | 68.4 | 77.3 | 45.4 | 64.4 | 75.2 | 63.4 | 76.9 | 60.1 |
| CLIP w/ 4 prompts | CLIP-Prim | 8.8 | 18.5 | 25.7 | 7.2 | 16.5 | 24.1 | 35.3 | 28.3 | 21.0 |
| | CLIP-Interv(Full) | 45.9 | 64.7 | 75.2 | 46.3 | 65.5 | 74.9 | 62.7 | 77.7 | 62.1 |
| | CLIP-Interv(Partial) | 49.7 | 70.4 | 77.4 | 50.1 | 69.8 | 78.3 | 66.2 | 79.5 | 65.2 |
| CLIP w/ 7 prompts | CLIP-Prim | 8.9 | 18.5 | 25.8 | 7.2 | 16.1 | 24.0 | 35.3 | 28.2 | 20.9 |
| | CLIP-Interv(Full) | 45.8 | 65.7 | 76.7 | 47.7 | 65.0 | 76.1 | 65.0 | 77.9 | 62.4 |
| | CLIP-Interv(Partial) | 51.5 | 72.1 | 80.7 | 51.1 | 70.1 | 80.2 | 67.3 | 79.8 | 65.4 |

Table A8: Results of ablation study about how scaling law impact CLIP variants to capture primitive concepts on CZSL task on MIT-States in the *closed-world* setting.

| Method | Training Prim GT | Evaluation Prim GT | Val AUC | | | Test AUC | | | Seen | Unseen | HM |
|---|---|---|---|---|---|---|---|---|---|---|---|
| | | | $k{=}1$ | 2 | 3 | 1 | 2 | 3 | | | |
| Oracle-Prim | ✓ | ✓ | 99.9 | 99.7 | 99.7 | 99.9 | 99.9 | 99.9 | 100.0 | 100.0 | 99.9 |
| CLIP-ResNet50-Prim | ✓ | ✓ | 6.1 | 12.9 | 19.2 | 4.0 | 10.5 | 15.9 | 28.8 | 21.1 | 15.4 |
| CLIP-ResNet50-Interv(Full) | ✓ | ✓ | 8.7 | 17.3 | 22.1 | 7.9 | 15.9 | 22.5 | 32.1 | 31.0 | 23.5 |
| CLIP-ResNet50-Interv(Partial) | ✓ | ✓✓ | 11.7 | 20.9 | 27.7 | 10.9 | 19.9 | 27.7 | 36.6 | 37.2 | 27.1 |
| CLIP-ResNet101-Prim | ✓ | ✓ | 6.6 | 14.1 | 20.4 | 4.1 | 10.3 | 15.9 | 28.9 | 20.7 | 15.6 |
| CLIP-ResNet101-Interv(Full) | ✓ | ✓ | 14.7 | 29.2 | 39.3 | 15.0 | 29.2 | 41.1 | 39.8 | 46.0 | 32.0 |
| CLIP-ResNet101-Interv(Partial) | ✓ | ✓✓ | 16.2 | 31.9 | 42.3 | 16.1 | 31.3 | 43.5 | 41.1 | 47.2 | 33.8 |
| CLIP-ViT-B/32-Prim | ✓ | ✓ | 8.2 | 17.0 | 24.1 | 6.9 | 15.6 | 22.8 | 34.0 | 27.9 | 20.4 |
| CLIP-ViT-B/32-Interv(Full) | ✓ | ✓ | 32.2 | 49.1 | 58.0 | 30.0 | 49.3 | 59.6 | 52.3 | 66.0 | 47.7 |
| CLIP-ViT-B/32-Interv(Partial) | ✓ | ✓✓ | 35.8 | 53.4 | 62.9 | 32.6 | 53.7 | 63.2 | 54.1 | 68.2 | 49.4 |
| CLIP-ViT-L/14-336px-Prim | ✓ | ✓ | 9.3 | 19.6 | 27.2 | 8.0 | 17.9 | 26.4 | 38.0 | 29.1 | 21.8 |
| CLIP-ViT-L/14-336px-Interv(Full) | ✓ | ✓ | 12.7 | 22.1 | 29.8 | 8.2 | 16.8 | 24.9 | 29.6 | 33.6 | 23.9 |
| CLIP-ViT-L/14-336px-Interv(Partial) | ✓ | ✓✓ | 16.3 | 27.6 | 37.0 | 11.1 | 23.7 | 33.5 | 36.7 | 37.3 | 27.5 |

interpretability of the composition models, reflecting that they do not capture primitive concepts better than the smaller models.

## A.6 Additional Experimental Details

### A.6.1 Baselines

We compare our approach for MIT-States against several state-of-the-art baselines in CZSL. LabelEmbed+ (LE+) Nagarajan & Grauman (2018) maps pair embeddings, combinations of word embeddings of attribute and object, and image embeddings into a joint semantic space using two separate MLP. Attribute as Operators (AOP) Nagarajan & Grauman (2018) regards attributes as linear transformations and the transformed object word embeddings are taken as composition embeddings. Task-Modular Modular Networks (TMN) Purushwalkam et al. (2019) trains a feature extraction model and a gating model which modifies

the classifier based on the given attribute-object pairs. SymNet Li et al. (2020) utilizes the symmetry properties of attribute-based transformation to learn object embeddings. Compositional Cosine Logits (Comp-Cos) Mancini et al. (2021) trains a linear layer as a composition function that projects a pair of attribute and object embeddings to a compositional space. Compositional Graph Embedding (CGE) Naeem et al. (2021) learns the dependency structure of attributes, objects, and their compositions using a Graph Convolutional Network (GCN) Kipf & Welling (2017). Since the proposed CGE is trained end-to-end and it finetunes its image encoder, we compare with the reported baseline, CompCos$_{\text{ff}}$, for MIT-States in closed-world setting, and with CompCos for MIT-States in open-world setting. Note that these models are task-specific supervised, whereas our approach only needs to learn composition models and is capable of performing the task in a zero-shot fashion.

We compare our CUB results to multiple state-of-the-art baselines. FEAT Ye et al. (2020) adapts instance embeddings to the target classification task with a set-to-set Transoformer function. DeepEMD Zhang et al. (2020) uses the Earth Mover's Distance as a metric to compute structural distances between dense image representations to determine image relevance during training. RENet Kang et al. (2021) combines self-correlational representation and cross-correlational attention to learn relational embeddings. S2M2 Mangla et al. (2020) regularizes feature manifolds via Manifold Mixup by focusing on learning a general purpose representation robust to changes in data distribution. PEM$_b$-NCM Hu et al. (2021) aims at processing feature vectors to be closer to Gaussian-like distributions to boost transfer learning accuracy.

### A.6.2 Metrics on MIT-States

Since a composition is trained only on seen composite concepts, the seen concepts can evaluate much better or worse than the unseen ones on validation and test splits. We follow the evaluation protocol from Purushwalkam et al. (2019) in order to reduce models' bias on seen compositions by applying calibration bias, which are scalars to be added to the predicted scores of unseen concepts. We vary the values of the calibration bias, and compute a list of accuracy scores of seen concepts and another list for unseen concepts. We compute a list of 20 bias scalars based on the difference between the predicted scores for seen and unseen composite concepts:

1. Obtain composition model's predicted probability distribution over seen and unseen concepts for each sample on test set.

2. For each unseen concept in the test set, compute the difference between the predicted score of most probable seen concept and the score of the ground truth concept. Denote this as $\Delta_{\text{diff}}$.

3. Sort $\Delta_{\text{diff}}$ and compute a list of 20 bias scalars by splitting $\Delta_{\text{diff}}$ into 20 chunks. Denote the list of 20 bias scalars as $\mathcal{B}$.

4. For each bias scalar $b$ in $\mathcal{B}$, add $b$ to the predicted scores of unseen concepts and evaluate model's performance. Therefore, we have a list of 20 accuracy scores of seen concept, and another list of 20 scores of unseen concept.

5. Compute the four metrics as introduced in Section 4.1.

