# OpenReview forum: "Do Vision-Language Pretrained Models Learn Composable Primitive Concepts?"
_TMLR — Accepted by TMLR_

### Review · Reviewer_ax67 · 2023-02-17

**Summary Of Contributions:**

The paper studies the ability of recent vision-language (VL) pretrained model to be able to capture primitive concepts associated with objects presented in an image such as colors, shapes and object’s attributes. It proposes a simple framework called Compositional Concept Mapping (CompMap) that tries to map the primitive concepts to more complicated composite concepts from a finite primitive concepts space. The authors evaluate three state-of-the-art VL models, CLIP, ViLT, and ALBEF, and measure the usefulness and interpretability of the learned composition models. The study shows that VL models can capture primitive concepts to some extent, but there is still room for improvement.

**Audience:**

No

**Claims And Evidence:**

No

**Requested Changes:**

1.	I would like to see the author’s clarification on their choices of evaluation metrics as mentioned in the weakness section.

2.	I also welcome further discussion on whether the proposed method can deal with complicated scenes in real-world data where multiple objects/humans or a good mix of objects of different kinds in one scene are present.

**Strengths And Weaknesses:**

Strengths:

1.	The paper is well motivated and the topic of study regarding machine capability to understand compositional concepts from low-level concepts is important.

2.	The proposed method is well explained and easy to follow.

Weaknesses:

1.	The paper uses different prompting to extract concepts out of an image but these would only work when the image contains single object or simple scenes. However, VL pretrained models are usually trained with more diverse set of data with complicated scenes. Their power is more about understanding visual scenes as a whole rather than extracting individual concepts.

2.	I doubt the effectiveness of the proposed model to formulate the problem as a classification task where labels are composite concepts. I also do not think that accuracy is a good metric to measure if the model understands these concepts well as it does not consider the similarity between target labels (e.g., “a white guy” and “a white man” are the same concept but two different labels).

3.	What are the total number of composite labels used in training? I guess the difference in performance between the Interv(Full) and Interv(Partial) are mainly because the large number of classes in the former case.

4.	The degradation in performance between “Prim” models and “Interv” models on CUB is interesting. Apart from what has explained in the paper, do you have any quantitative or qualitative analysis to explain this?

---

> ### Author Response · Authors · 2023-02-25
> **Response to Reviewer ax67 (1/2)**
>
> Thank you for your helpful comments and critical questions. Here, we summarize your major concerns/questions and incorporate your comments into our manuscript.
>
> **Q1. The paper uses different prompting to extract concepts out of an image but these would only work when the image contains single object or simple scenes. However, VL pretrained models are usually trained with more diverse set of data with complicated scenes. Their power is more about understanding visual scenes as a whole rather than extracting individual concepts.**
>
> A1. Respectfully, we want to push back against this as a weakness. The goal of our paper is to understand more about _how_ VL models work under the hood, and thus the focus on simple scenes is a deliberate choice in this study, not a weakness. We agree that these neural network models are trained to label more complex scenes. However, it is not known exactly how they do so. There are several possibilities:  they may understand the scenes by either (1) matching prototypes of the scenes (i.e., general appearance of basketball court),  (2) decomposing the scenes into constituent parts or “primitives” (e.g., baskets, bleachers, wooden floors), or a combination of both. While CLIP and most of the recent VL models have the former as their pre-training objective, namely directly matching images with complicated scene descriptions, it is important and meaningful for neural networks to understand the scenes by decomposing scenes to their primitives. There are many reasons we might prefer an AI system which reasons about wholes by reasoning about the parts. For example, it improves trust. Human users who see a model knows what a “red bird” is will assume the model knows what “red” is, and will be confused and frustrated if the model does not. Moreover, as we mentioned in our Introduction section, reasoning over parts provides better interpretability and potential compositional generalizability. This allows neural networks to reuse the learned primitive concepts for various complex concepts, instead of memorizing the prototypes of different scenes during pretraining. Our paper aims at investigating whether primitive concepts emerge automatically from the existing VL pre-training objectives.
>
>
> **Q2. I doubt the effectiveness of the proposed model to formulate the problem as a classification task where labels are composite concepts. I also do not think that accuracy is a good metric to measure if the model understands these concepts well as it does not consider the similarity between target labels (e.g., “a white guy” and “a white man” are the same concept but two different labels).**
>
> A2. MIT-States and CUB both have expert-designed concept vocabularies and do not have semantically similar concepts, such as “guy” and “man”. We agree that a standalone accuracy metric does not show if the VL pretrained models understand primitive concepts well. This is exactly the motivation why we propose the “interpretability” metric as well, which is the performance difference between an oracle model trained on ground truth primitive concepts and a composition model learned from predicted concept activations from VL pretrained models. Our interpretability metric is metric-agnostic – it can work with any evaluation metrics, such as accuracy and area-under-curve (AUC in MIT-States).
>
>
> **Q3(a). What are the total number of composite labels used in training?**
>
> A3(a). In Section 4.1 - Datasets, we mentioned that, in MIT-States, there are 1262 composite labels in training set, and 800 composite labels (400 seen + 400 unseen) in test set. In CUB, there are 200 bird classes in total, and the number of classes used during training depends on the n-way k-shot few-shot learning task.
>
>
> **Q3(b). I guess the difference in performance between the Interv(Full) and Interv(Partial) is mainly because the large number of classes in the former case.**
>
> A3(b). Interv(Full) means to pass in the binary primitive concept activations to composition models, and Interv(Partial) means to only set the correct primitive concept to 1 in the predicted concept activations. The different performance between Interv(Full) and Interv(Partial) reflects that the learned composition models capture spurious correlations from the predicted activations of irrelevant primitive concepts.

---

> > ### Author Response · Authors · 2023-02-25
> > **Response to Reviewer ax67**
> >
> > **Q4. The degradation in performance between “Prim” models and “Interv” models on CUB is interesting. Apart from what has explained in the paper, do you have any quantitative or qualitative analysis to explain this?**
> >
> > A4. In Section 5.2 (Table 4 and Figure 3), we observed that CLIP-Prim usually incorrectly correlates the bird attributes with the bird species. This leads to the performance degradation between “Prim” models and “Interv” models on CUB.
> >
> > In addition, we also conducted an ablation study to investigate the effectiveness of concept prompting by replacing concept prompting with random projection layer or learned projection layer (Table 6 and Section 5.3). We found that CLIP-Prim only moderately outperforms the learned and random projection baselines, indicating that concept prompting can be viewed as some projection of the CLIP visual embeddings. During evaluation on CUB, more information is removed in Full intervention, leading to the performance degradation.
> >
> > We welcome suggestions from reviewers for additional analyses.
> >
> >
> > **Q5. I also welcome further discussion on whether the proposed method can deal with complicated scenes in real-world data where multiple objects/humans or a good mix of objects of different kinds in one scene are present.**
> >
> > A5. Our proposed pipeline CompMap is agnostic to image- or region-based pretrained models, as long as the model provides a vision-language interface (as in CLIP). There are existing VL pretrained models that are specifically designed for complex scenes and multiple objects, such as GLIP [1], Region-CLIP [2]. We could readily apply our method to these models if we were interested in understanding if/how models learn to decompose more complex scenes into primitive concepts. Given the negative results on the simple scenes we investigate here (e.g., failing to distill a stable concept “ripe” despite reliably labeling “ripe apple”, “ripe banana”, “ripe berry”, etc), it is reasonable to assume that models will also fail to decompose more complex scenes which likely have more abstract primitives. However, indeed, models with different inductive biases, like those above, might perform better than the models we test here. This is a very interesting direction for future work.
> >
> >
> > [1] Grounded Language-Image Pre-training. CVPR 2022.
> >
> > [2] RegionCLIP: Region-based Language-Image Pretraining. CVPR 2022.

---

### Review · Reviewer_ULW2 · 2023-02-17

**Summary Of Contributions:**

The paper presents a novel framework called Compositional Concept Mapping (CompMap) to evaluate the effectiveness of visual-linguistic (VL) pre-trained models in learning primitive concepts. By measuring the performance of composition model learning, CompMap enables a quantitative assessment of the quality of learned concepts. The authors conduct extensive experiments using a variety of state-of-the-art VL pre-trained models and provide both quantitative and qualitative analysis to reveal that these models lack interpretability, suggesting that they do not learn primitive concepts in a human-understandable way. However, the models still exhibit high performance in visual recognition tasks. The authors highlight the need to improve VL pretraining methods by emphasizing the importance of learning interpretable primitive concepts. The paper provides valuable insights and a useful framework for evaluating VL pretraining, and the authors make their code and models publicly available for further research.

**Audience:**

Yes

**Broader Impact Concerns:**

No concern.

**Claims And Evidence:**

Yes

**Requested Changes:**

1. The relationship between the interpretability and the score measured by model intervention is not clear to the reviewer (and potentially to the further readers). Thus the reviewer requests a detailed illustration of such relationship. The illustration can be formulated by describing why a small difference can lead to a highly interpretable compositional model, and why a highly interpretable compositional model can be related to the paper's intention on judging the VL model's primitive concepts. It would be great to have a complete step-by-step illustration where no step has confusion to the reader.  More details: for "why a small difference can lead to a highly interpretable compositional model", the reviewer expect a explanation on why a small difference can not lead to a low interpretable model as well.
2. For the evaluation metrics in Sec 4.1, provide a concrete calculation in Appendix. This will help the reproduction of this paper.
3. [Minor] In Sec 4.2., why the models from different VL model take different ViT backbone? Also, it would be great to also show the trend of usefulness and interpretability as the model size grows. This might be outside the scope of this paper, but the
4. In Table 4, it would be better to give a break down metric of the accuracy. I think that the current model only considers correct if the whole set of primitive concepts match?
5. [Minor] Any reasons on why CLIP-Prim has a lower unseen score but a higher seen score? Since one of the goal of incorporating primitive concepts is to improve the generalization.
6. In Appendix A.1, the CLIP model takes dot product (actually cosine similarity since the vectors are normalized?) and other model takes matching score. Thus the CLIP activations can be negative while the other two models' output is always positive. Would it affect the comparison of these VL models?

**Strengths And Weaknesses:**

Strengths:
- The paper proposes a novel framework called Compositional Concept Mapping (CompMap) to evaluate the effectiveness of visual-linguistic (VL) pre-trained models in learning primitive concepts. The proposed framework enables a quantitative assessment of learned concepts by measuring the performance of composition model learning.
- The paper proposes the evaluation method of the usefulness and the interpretability.
- The authors conduct extensive experiments using a range of recent VL pre-trained models, and provide both quantitative and qualitative analysis to reveal that the models lack interpretability, indicating that they do not learn primitive concepts in an understandable way.
- The authors make their code and models publicly available, which can facilitate further research and development in this field. The paper also provides valuable insights into the limitations of existing VL pre-trained models and suggests potential directions for future research.

Weakness:
- The relationship between the interpretability metric (i.e., the difference of two models on GT input) and the interpretability of composition model is not clear given the paper. I can get a sense of such relationship but not confident enough, but I think that it is more like a writing concern (although very important for the paper presentation).

---

> ### Author Response · Authors · 2023-02-25
> **Response to Reviewer ULW2**
>
> Thank you for your helpful comments and constructive suggestions. Here, we summarize your major concerns/questions and incorporate your comments into our manuscript.
>
> **Q1. The relationship between the interpretability and the score measured by model intervention is not clear. Thus the reviewer requests a detailed illustration of such relationship. The illustration can be formulated by describing why a small difference can lead to a highly interpretable compositional model, and why a highly interpretable compositional model can be related to the paper's intention on judging the VL model's primitive concepts. It would be great to have a complete step-by-step illustration where no step has confusion to the reader. More details: for "why a small difference can lead to a highly interpretable compositional model", the reviewer expects an explanation on why a small difference can not lead to a low interpretable model as well.**
>
> A1. In our work, we assume that the Oracle-Prim would have (nearly) perfect performance, which we verified empirically. Therefore, we use the absolute difference between Oracle-Prim and Interv(Full) results. When the performance difference is small, then it reflects that the VL pretrained models learn primitive concepts well. In order to make it more comprehensive, we can normalize the “interpretability” by the performance of Oracle-Prim.
>
> We have a step-by-step illustration in Figure 2 to demonstrate how interpretability is computed. The intuition is that the smaller the performance gap is, the more interpretable the concept activations are. When a VL model predicts interpretable concept activations, we then consider this model as a model that learns primitive concepts. We will include a step-by-step illustration to elaborate how the interpretability metric reflects how well a VL pretrained model learns primitive concepts in our revision.
>
>
> **Q2. For the evaluation metrics in Sec 4.1, provide a concrete calculation in Appendix. This will help the reproduction of this paper.**
>
> A2. Thank you for the suggestion. We will incorporate a section introducing how we compute the evaluation metrics in Appendix.
>
>
> **Q3. [Minor] In Sec 4.2., why the models from different VL model take different ViT backbone? Also, it would be great to also show the trend of usefulness and interpretability as the model size grows. This might be outside the scope of this paper.**
>
> A3. The reason is that ViLT only releases checkpoints with ViT-B/32, and ALBEF only releases checkpoints with ViT-B/16. However, this would not impact the observations of the paper, because the comparisons are intra-model (e,g,. CLIP-Prim compares to CLIP-Interv(Full), but not to ALBEF-Prim).
>
> An ablation study of scaling up VL models is a great suggestion. We are working on this ablation study with CLIP, since CLIP shares the checkpoints with different visual backbones.
>
>
> **Q4. In Table 4, it would be better to give a breakdown metric of the accuracy. I think that the current model only considers correct if the whole set of primitive concepts match?**
>
> A4. In the paper, we treat the correlations between primitive concepts and composite concepts as independent. For example, "red apple" has two positive primitive concepts, "red" and "apple". If the model correlates "red" correctly, but "apple" incorrectly, then accuracy is 50%. We will involve the results of matching the whole set of primitive concepts in the revision.
>
>
> **Q5. [Minor] Any reasons on why CLIP-Prim has a lower unseen score but a higher seen score? Since one of the goal of incorporating primitive concepts is to improve the generalization.**
>
> A5. This shows the use of primitive concepts does not improve generalization as much as we hoped, which is in line with our findings that the VL pretrained models do not learn primitive concepts well.
>
>
> **Q6. In Appendix A.1, the CLIP model takes dot product (actually cosine similarity since the vectors are normalized?) and other model takes matching score. Thus the CLIP activations can be negative while the other two models' output is always positive. Would it affect the comparison of these VL models?**
>
> A6. CLIP takes dot product over the normalized text embeddings and normalized image embeddings. ViLT and ALBEF take the matching scores, which are logits. Thus, they can all be positive or negative values in principle. We will clarify this in our revision.

---

> > ### Comment · Reviewer_ULW2 · 2023-03-18
> > **Discussions**
> >
> > I think that the claim
> > ```
> > When the performance difference is small, then it reflects that the VL pretrained models learn primitive concepts well.
> > ```
> > is not conclusive to me.
> >
> > Here is my example to illustrate what I thought:
> >
> > ----
> > Suppose that we just have three colors red, black, and blue. Also, we have two classes of birds: red bird and black bird (blue is just a random color which never appear in GT, but the visual model can predict it). We also have two visual models (A and B) to compare their interpretability.
> >
> > For model A, it predicts well-calibrated and accurate value (e.g., for red bird, it predicts {red: 1., black:0., blue:0}).
> >
> > For model B, it predicts correct label but with a very small value 0.01 and always predicts the blue color with high value (e.g., for red bird, it predicts {red:0.01, black:0.0, blue:1.}).
> >
> > Then we learned the classifier ("Learned Composition Model" in paper) for model A and model B. Model A and B both get the perfect classifier ({class red: {red:1, black:0}, class balck: {red:0, black:1}), and thus get the same "intervening" score.
> >
> > Thus, in this example, we get two models A and B with the same intervening score but has different interpretability to me. Having this "same score with different interpretability", we can twist this example to get "lower score with better interpretability" and "higher score with worse interpretability".
> >
> > Of course, this is an extreme case but can be extensible to several ways. E.g., we can change the  correct prediction probability of model A and B (e.g., model A predicts {red: 1., black:0., blue:0} for 80%, model B predicts {red:0.01, black:0.0, blue:1.} for 80% as well), and get an intervening score different to 100% (e.g., the above example is 80%).
> >
> > -----
> >
> > I want to use this example to spotlight two questions of the assumption "higher intervention performance (since usually GT score is the highest, thus smaller gap means higher score) can always lead to better interpretability":
> > 1. This deduction is not natural and needs more proof in the work. We can not simply rely on "GT gets best score" to get the conclusion "better model gets better score".
> > 2. The term "interpretability of primitive concepts" might need a more concrete definition.

---

> > > ### Author Response · Authors · 2023-03-23
> > > **Response to Reviewer ULW2**
> > >
> > > Thank you for sharing your valuable insights on our "interpretability" metric!
> > > We would also like to take the opportunity to clarify our definition of interpretability, and emphasize that the conclusions (that VL pretrained models do not learn primitive concepts well) still hold.
> > >
> > > (1) The focus of interpretability metric is to reflect how interpretable the composition models are. This is also how we define interpretability metric in Section 3.2 “Quantifying interpretability”. The logic behind this metric is that the full intervention reflects whether a composite concept is correlated with the correct set of primitive concepts. Therefore, a large performance gap indicates a composition model is not interpretable (i.e., not rely on irrelevant primitive concepts for the composite concepts); and a small gap means the composition model is interpretable (i.e., correlate correct primitives for the composite concepts).
> > >
> > > Even though a composition model is interpretable, it does not mean that the learned weights are similar to the composition models learned from ground truth primitives. Therefore, we conducted weight accuracy experiments (Section 5.2) and the results are complementary to our observations (Table 2) that composition models learned from VL models are not interpretable.
> > >
> > > Intuitively, the interpretability of composition models is correlated with primitive concept interpretability, but as the reviewer pointed out, a small gap according to the interpretability metric may not be aligned with the intuitive definition of interpretability, when the concept activations are mis-calibrated. We will clarify this limitation in the revision. However, our conclusion that VL models do not learn interpretable primitive concepts still holds, as a large gap indeed implies that the primitive concept activations are not interpretable, regardless of the calibration issue.
> > >
> > > (2) “Interpretability of primitive concepts” refers to how well the VL-model’s predicted primitive concept activations reflect the properties of binary ground truth primitives.
> > >
> > > For example, if there are three concepts [red, blue, orange], then the interpretable representation of concept “red” is {red: 1, blue:0, orange:0}. Another representation of “red” is {red: 0.8, blue: 0.1, orange: 0.1}, which is less interpretable than the former one.

---

> > > > ### Comment · Reviewer_ULW2 · 2023-03-28
> > > > **About full intervention**
> > > >
> > > > Thanks for the detailed explanation. Would you please expand more on the statement "full intervention reflects whether a composite concept is correlated with the correct set of primitive concepts"? I still can not truly convince myself this relationship by following the formulation of "full intervention" and reading the descriptions in paper/discussions. Any pointers or reference are recommended as well.
> > > >
> > > > Btw, the discussion period of paper had ended and all reviewers should have given the paper recommendation. Overall, I am lean positive with this paper, and I just want to understand more about the philosophy behind this paper.

---

### Review · Reviewer_qz9s · 2023-02-18

**Summary Of Contributions:**

In this paper, the authors study the problem of whether pretrained vision language models can extract primitive concepts and whether the extracted concepts can used to train a composition model in a more interpretable manner. Various studies are reported to empirically study three vision-language models' behavior.

**Audience:**

Yes

**Broader Impact Concerns:**

This is not really a concern but more like a question. Since the authors mentioned potential bias research on VL, could the dataset be further extend to datasets with racial or gender annotations to understand the potential bias issue of VL models? Or maybe just given existing datasets, could the authors further discuss on the possible biased concepts in the VL models?

**Claims And Evidence:**

No

**Requested Changes:**

Please check the cons for details.

**Strengths And Weaknesses:**

Pros:
1. The problem studies here is of high research value.
2. The reported results are interesting.
3. The presentation is overall clear.

Cons:
1. Currently, the experiment setup could use further improvement.
- a. The concepts could be really fine-grained in a very tiny region of the whole image. Therefore, it is possible this is the bottleneck limiting existing models. Comparing text embeddings of prompts and visual embeddings of spatial resolution could provide further justification and understanding the model's behavior.
- b. The models may interpret the concepts in a different manner than us. Therefore, it is also necessary to check the spatial heat-map to understand whether models really localize to the correct region for a concept or maybe they have their own "definition" of the concept by checking if there is a pattern in terms of highly activated regions.
- c. The current conclusion is that the learned concepts of VL models are not that interpretable. This raise interesting combination and comparison with recent work [1,2]. [1] shows that some learned latent textual concepts could be of more help for the downstream tasks and [2] shows that pretrained language model could be used to generate the helpful query to prompt VL model for finegrained recognition. Therefore, it would be very interesting to show and discuss: 1) if the concepts of VL models should be better not interpretable, could learned latent concept further improve the performance? 2) if the concepts could be actually better retrieved by better query design, can the language model help in this scenario?


[1] Visual Classification via Description from Large Language Models, ICLR 23

[2] Learning to Decompose Visual Features with Latent Textual Prompts, ICLR 23

---

> ### Author Response · Authors · 2023-02-25
> **Response to Reviewer qz9s (1/2)**
>
> Thank you for your helpful comments and critical questions. Here, we summarize your major concerns/questions and incorporate your comments into our manuscript.
>
> **Q1. The concepts could be really fine-grained in a very tiny region of the whole image. Therefore, it is possible this is the bottleneck limiting existing models. Comparing text embeddings of prompts and visual embeddings of spatial resolution could provide further justification and understanding the model's behavior.**
>
> A1. This is a good point you raise, we agree that the primitive concepts could be really fine-grained in a very tiny region. To investigate your concern, we computed a quantitative analysis (similar to Table 4) on CUB dataset to study the relationship between the region size of the concepts and the correlation between primitives and composite concepts. We computed the accuracy that measures whether the most activated weights align with the ground truth primitive concepts:
>
> For bird attribute “crown color”, CLIP-Prim reaches accuracy of 52.1% (vs. random baseline at 6.7%).
>
> For “upper tail color”, CLIP-Prim reaches 57.4%  (vs. random baseline at 6.7%).
>
> For “leg color”, CLIP-Prim reaches 16.7%  (vs. random baseline at 6.7%).
>
> For “breast color”, CLIP-Prim reaches 50.0%  (vs. random baseline at 6.7%).
>
> For “upperparts color”, CLIP-Prim reaches 46.5%  (vs. random baseline at 6.7%).
>
> For “back color”, CLIP-Prim reaches 56.5%  (vs. random baseline at 6.7%).
>
>
> Based on the above preliminary results, we observe no strong correlation between the region size (or spatial resolution) of the primitive concepts and how well they are recognized by the Vl pretrained models. As in our response to ULW2, we acknowledge that using region-based VL models may help us better capture the primitive concepts in tiny regions.
>
>
> **Q2. The models may interpret the concepts in a different manner than us. Therefore, it is also necessary to check the spatial heat-map to understand whether models really localize to the correct region for a concept or maybe they have their own "definition" of the concept by checking if there is a pattern in terms of highly activated regions.**
>
> A2. In this paper, we are interested in whether models learn a decomposition of concepts that is consistent with humans’ intuitive decomposition. We agree with you that models might decompose concepts differently (e.g., rather than using one stable “red” concept which corresponds to the word “red” and is used across many bird species and body parts, a model might learn many different “red” concepts, one for each species, and represent them as entirely unrelated). If that is the case, it may appear interpretable using heat maps, but  in our paper, it should still achieve a low interpretability score, since this is not aligned with human intuitions about the concept “red”. Characterizing the models unhumanlike concepts is an interesting question, but is not the goal of this paper. Thus, we acknowledge the reviewer's suggestion about the spatial interpretation as  relevant, but this would require additional ground truth annotations of the regions of primitive concepts and would answer a question that is different from the primary one we lay out. Therefore, we consider the spatial interpretation as a followup metric for our interpretability metric. As we demonstrated in the paper, existing VL models fail on our interpretability metric, so it may not be meaningful to quantitatively evaluate the spatial localization of the concepts.

---

> > ### Author Response · Authors · 2023-02-25
> > **Response to Reviewer qz9s (2/2)**
> >
> > **Q3. The current conclusion is that the learned concepts of VL models are not that interpretable. This raise interesting combination and comparison with recent work [1,2]. [1] shows that some learned latent textual concepts could be of more help for the downstream tasks and [2] shows that pretrained language model could be used to generate the helpful query to prompt VL model for finegrained recognition. Therefore, it would be very interesting to show and discuss: 1) if the concepts of VL models should be better not interpretable, could learned latent concept further improve the performance? 2) if the concepts could be actually better retrieved by better query design, can the language model help in this scenario?**
> >
> > A3. These are interesting papers that we will include in our discussion of related work. However, they address a different question than what we address here. Learning latent concepts could bring better performance, but such methods usually require the models to learn task-specific trainable tokens. This actually demonstrates the importance for VL pretrained models to learn primitive concepts, which is our motivation to study whether primitive concepts automatically emerge during the VL pretraining. If VL pretrained models indeed learn primitive concepts, these models enable (possibly) compositional generalization, better human-model interaction, and understanding of models’ predictions, which cannot be handled by latent concepts without additional annotated data.
> >
> > Also, we explored different query designs in Appendix A.3, and the observation aligns with our results in the main paper. However, we do agree that a better backbone of language model and careful design of queries will be important.
> >
> >
> > [1] Visual Classification via Description from Large Language Models. ICLR 2023.
> >
> > [2] Learning to Decompose Visual Features with Latent Textual Prompts. ICLR 2023.
> >
> > [3] Concept Bottleneck Models. PMLR 2020.
> >
> > [4] Do Concept Bottleneck Models Learn as Intended? ICLR 2021.

---

> > > ### Author Response · Authors · 2023-03-05
> > > **Response to Reviewer qz9s about Broader Impact Concerns**
> > >
> > > We agree that bias research is a great direction, but it will be a stretch for our proposed method (CompMap), since biases are not inherently compositional. As for the datasets we used, the concepts are objective visual concepts (e.g., colors, shapes, etc.), so there are no concerns about biased concepts.

---

### Author Response · Authors · 2023-03-05
**General Response**

Dear Editors and Reviewers,

Many thanks for your constructive comments that help improve our manuscript. We have revised the manuscript to address your comments.

To clarify the proposed CompMap method and metrics of our work, we referred to our Figure 2 in Section 3.2 to clarify how interpretability metric works. We also added details about how we compute performance metrics on MIT-States to improve the reproducibility of our work (Appendix A.5.2). In terms of experiments, we included additional results required by reviewers to investigate the relationship between the region size of primitive concepts and the correlations between the primitives and composite concepts (Section 5.2). Plus, we conducted an ablation study to investigate the scaling law impact on how well VL pretrained models can capture primitive concepts (Appendix A.4). Plus,  More analysis and elaboration have been done to shape our contributions better.

Hope this work could receive your full consideration of acceptance for publication. Thank you!

Best,
Authors

---

### Decision · Action_Editors · 2023-03-28

**Recommendation:** Accept as is

**Comment:**

This paper presents a framework for measuring how well vision-language pre-trained models (such as CLIP, ViLT and ALBEF) can learn primitive concepts, in terms of usefulness and interpretability. After author response, it received two Leaning Accept and one Leaning Reject recommendations, so it is really a borderline case.

On one hand, all the reviewers agree that the research question raised by this paper is important and it will be of interest for those who would like to understand how pre-trained VL models actually work under the hood. The paper is overall well written, and provides new insights and quantitative assessment for evaluating the effectiveness of VL models in learning primitive concepts.

On the other hand, due to the nature that this paper is an analysis paper and aims to provide new quantitative assessment to inspect the current VL models to learn primitive concepts, reviewers have commented that some additional clarification is needed and the paper needs to put more evidence and explanation of the new concepts into writing, such as the definition and measurement of "interpretability" in the paper. Reviewer qz9s also commented that additional experiments and analysis can be added, such as further analysis between region size and the attribute recognition results.

Overall, the editor thinks that the authors have done a relatively good job at rebuttal. In many cases, we care much about pushing the SoTA and here in this paper, the authors take a step back and try to understand better about the current VL models. Surely, the analysis provided in this paper can be further enhanced, but overall, the editor thinks that the merits slightly outweigh the flaws of the paper, and would like to recommend acceptance of the paper. The editor encourages the authors to take all the reviewers' feedback into consideration to prepare the camera-ready version.

**Audience:**

Audience who are interested in understanding and better probing the current vision-language pre-trained models such as CLIP may be interested in this paper.

**Claims And Evidence:**

This paper is an analysis paper that tries to understand better about the current VL pre-trained models. Due to several new concepts are introduced, it is hard to say all the claims are clearly supported by convincing evidence. However, overall, this paper did a good job in making claims.